# NADPH Oxidase Subunit CYBB Confers Chemotherapy and Ferroptosis Resistance in Mesenchymal Glioblastoma via Nrf2/SOD2 Modulation

**DOI:** 10.3390/ijms24097706

**Published:** 2023-04-22

**Authors:** I-Chang Su, Yu-Kai Su, Syahru Agung Setiawan, Vijesh Kumar Yadav, Iat-Hang Fong, Chi-Tai Yeh, Chien-Min Lin, Heng-Wei Liu

**Affiliations:** 1Graduate Institute of Clinical Medicine, College of Medicine, Taipei Medical University, Taipei City 11031, Taiwan; ichangsu@gmail.com (I.-C.S.); yukai.su@gmail.com (Y.-K.S.); 18149@s.tmu.edu.tw (I.-H.F.); 2Department of Neurology, School of Medicine, College of Medicine, Taipei Medical University, Taipei City 11031, Taiwan; 3Division of Neurosurgery, Department of Surgery, Taipei Medical University-Shuang Ho Hospital, New Taipei City 23561, Taiwan; 4Taipei Neuroscience Institute, Taipei Medical University, Taipei City 11031, Taiwan; 5International Ph.D. Program in Medicine, College of Medicine, Taipei Medical University, Taipei City 11031, Taiwan; setiawan.syahru@gmail.com; 6Department of Medical Research & Education, Taipei Medical University-Shuang Ho Hospital, New Taipei City 23561, Taiwan; vijeshp2@gmail.com (V.K.Y.); ctyeh@s.tmu.edu.tw (C.-T.Y.); 7Continuing Education Program of Food Biotechnology Applications, College of Science and Engineering, National Taitung University, Taitung 95092, Taiwan

**Keywords:** CYBB, mesenchymal, glioblastoma, temozolomide, ferroptosis, SOD2, erastin

## Abstract

Glioblastoma multiforme (GBM) is a highly heterogeneous disease with a mesenchymal subtype tending to exhibit more aggressive and multitherapy-resistant features. Glioblastoma stem-cells derived from mesenchymal cells are reliant on iron supply, accumulated with high reactive oxygen species (ROS), and susceptible to ferroptosis. Temozolomide (TMZ) treatment is the mainstay drug for GBM despite the rapid development of resistance in mesenchymal GBM. The main interconnection between mesenchymal features, TMZ resistance, and ferroptosis are poorly understood. Herein, we demonstrated that a subunit of NADPH oxidase, CYBB, orchestrated mesenchymal shift and promoted TMZ resistance by modulating the anti-ferroptosis circuitry Nrf2/SOD2 axis. Public transcriptomic data re-analysis found that CYBB and SOD2 were highly upregulated in the mesenchymal subtype of GBM. Accordingly, our GBM cohort confirmed a high expression of CYBB in the GBM tumor and was associated with mesenchymal features and poor clinical outcome. An in vitro study demonstrated that TMZ-resistant GBM cells displayed mesenchymal and stemness features while remaining resilient to erastin-mediated ferroptosis by activating the CYBB/Nrf2/SOD2 axis. The CYBB maintained a high ROS state to sustain the mesenchymal phenotype, TMZ resistance, and reduced erastin sensitivity. Mechanistically, CYBB interacted with Nrf2 and consequently regulated SOD2 transcription. Compensatory antioxidant SOD2 essentially protected against the deleterious effect of high ROS while attenuating ferroptosis in TMZ-resistant cells. An animal study highlighted the protective role of SOD2 to mitigate erastin-triggered ferroptosis and tolerate oxidative stress burden in mice harboring TMZ-resistant GBM cell xenografts. Therefore, CYBB captured ferroptosis resilience in mesenchymal GBM. The downstream compensatory activity of CYBB via the Nrf2/SOD2 axis is exploitable through erastin-induced ferroptosis to overcome TMZ resistance.

## 1. Introduction

Glioblastoma multiforme (GBM) is a highly heterogeneous disease and is among the most common and aggressive forms of primary brain cancer. The molecular context of GBM has been categorized and successfully classified using multiple omics approaches. For instance, a study on The Cancer Genome Atlas (TCGA) published in 2010 discovered four subtypes of GBM: proneural, neural, classical, and mesenchymal. These subtypes are highly correlated with genetic modifications, such as tumor protein 53 (*TP53*), epidermal growth factor receptor (*EGFR*), and neurofibromatosis type 1 (*NF1*) mutations [1,2]. GBM tumors with the mesenchymal signature have received considerable attention because they are more aggressive than those with other transcriptomic characteristics. Approximately a third of patients have the mesenchymal subtype of GBM, and these patients have a poor outcome. Compared with other subtypes of GBM tumors, mesenchymal tumors display higher levels of aggressiveness, angiogenic activity, hypoxia, inflammation, and multitherapy resistance [3]. Thus, determining both standard-of-care and treatment approaches for patients with mesenchymal GBM is challenging. The identification of the key regulators of mesenchymal GBM and their downstream signaling pathways can enhance our understanding regarding this intricate transcriptional regulatory network and facilitate the development of effective treatment strategies.

GBM remains among the most fatal and difficult malignancies to treat, and less than 5% of patients survive for more than 5 years. Since 2005, patients with GBM undergoing maximally safe surgical resection have received temozolomide (TMZ), an alkylating drug approved by the Food and Drug Administration. In addition to causing DNA alkylation, TMZ causes oxidative stress, which is lethal for cancer cells [4]. Although reactive oxygen species (ROS) are crucial second messengers in intracellular signaling pathways that moderately increase the oncogenic phenotype of cancer cells, they play a conflicting dual role. Excess ROS can damage proteins, lipids, and DNA, resulting in both apoptosis and ferroptosis [5]. Furthermore, TMZ increases antioxidative activity, DNA repair mechanisms, and mitochondrial coupling, which promote tumor growth and chemoresistance [6]. GBM recurrence is associated with shorter patient survival, resistance to TMZ, and increased Nuclear factor E2-related factor 2 (Nrf2)-targeted antioxidant system expression. Several mitochondria-related genes were differentially expressed between parental and resistant GBM cells, and among them, the superoxide dismutase 2 (*SOD2*) gene substantially affected various factors, including cancer stemness, treatment resistance, and patient survival [7]. *SOD2* may regulate and neutralize a substantial portion of superoxide generated by the NADPH oxidase (NOX) family [8]. The NOX family plays a protumorigenic role and promotes the mesenchymal features and carcinogenesis of several cancer cells [9,10]. Resistance to TMZ can be overcome by employing various approaches. Identifying the aberrant antioxidant system and treating it with pro-oxidants may be an effective strategy.

Ferroptosis is a form of regulated cell death governed by excess iron and characterized by ROS accumulation and lipid peroxidation, which cause membrane rupture. This nonapoptotic form of cell death was first described in 2012 by a study employing erastin to induce ferroptosis [11]. Ferroptosis is effective in eliminating treatment-resistant tumor cells. Moreover, ferroptosis has been linked with drug resistance upon TMZ treatment. TMZ treatment has been known to accumulate mitochondrial and lipid ROS, increase labile iron pool, while depleting antioxidative capacity in glioma cells [12,13], thereby indicating ferroptosis induction as an alternative TMZ mechanism of action in glioblastoma. On the other hand, resistance to TMZ also closely connects with the way how glioma cells defend against the deleterious impact of a high ROS state and ferroptosis resilience [7,14]. Establishing this link may enable the use of ferroptosis inducers to treat gliomas. Nrf2 inhibition increased the sensitivity of glioma cells to TMZ [15]. In addition, TMZ treatment promoted the generation of glutathione to confer resistance [16]. Originally described as a multityrosine kinase inhibitor, sorafenib triggered ferroptosis and altered TMZ sensitivity which could be reversed through ROS elimination [17]. These findings indicate that excessive ROS generation or pro-oxidant treatment may perturb TMZ sensitivity.

The identification of the possible link between chemoresistance and ferroptosis susceptibility, particularly in mesenchymal GBM, is helpful in determining the potential of ferroptosis to eliminate TMZ resistance. The findings of this study demonstrated that Cytochrome B-245 Beta Chain (*CYBB*), a major catalytic subunit of NOX, contributed to resistance upon TMZ treatment and altered the sensitivity of mesenchymal GBM to ferroptosis. Furthermore, aberrant CYBB activation in mesenchymal GBM led to the accumulation of mitochondrial ROS and activation of the compensatory Nrf2/SOD2 circuitry. This study determined the susceptibility of mesenchymal GBM to ferroptosis for overcoming TMZ resistance.

## 2. Results

### 2.1. CYBB Linked Mesenchymal Signatures to Treatment Resistance and Nr2 Activation

GBMs are aggressive brain tumors characterized by extensive intertumor and intratumor heterogeneity. To investigate cellular and molecular heterogeneity in GBM model systems, we analyzed a previously reported single-cell RNA sequencing dataset from the Chinese Glioma Genome Atlas (CGGA). Transcriptomic signatures of mesenchymal cells, treatment resistance, and the Nrf2 signaling pathway, a well-known intrinsic defense mechanism against oxidative stress, were explored. In this dataset, cells originating from tumor components were more abundant than those from peritumoral or normal counterparts and clustered into 14 subgroups (Figure 1A). Individual t-SNE plots were constructed to illustrate the expression levels of our genes of interest, namely *CYBB*, *NFE2L2* (*Nrf2*), and *SOD2*, in each cluster; the localization of CYBB expression was noted in clusters numbered 5, 7, 8, and 13 (Figure 1B). Relevant gene signatures in this GBM study, including Verhaak_GBM_Mesenchymal, Glioma_stemness, Phillips_GBM_Proliferative, WP_Nrf2_pathway, Anastassiou_Invasiveness, and Segerman_Resistance, were counted and displayed in each t-SNE plot. CYBB expression was consistently observed in cell clusters associated with highly activated mesenchymal GBM, Nrf2 pathway activation, and treatment resistance signatures (Figure 1C). In addition, a comparative analysis of gene signatures indicated that the mesenchymal phenotype, Nrf2 signaling, and treatment resistance were highly activated in previously defined clusters with high CYBB expression, including clusters numbered 7, 8, and 13 (Figure 1E). Moreover, mesenchymal signature activation was positively correlated with treatment resistance and Nrf2 activation, suggesting the interconnection of mesenchymal subtypes with other crucial pathways (Figure 1D). Therefore, single-cell profiling data indicated that the high expression of CYBB reflected mesenchymal signature activation in heterogeneous GBM tumor cells.

To confirm the intratumor heterogeneity observed in tumor cells at the single-cell level, on the basis of high CYBB expression observed in mesenchymal cells, we examined the transcriptomic profiles of bulk tumor cells. We analyzed the TCGA GBM dataset and determined that CYBB and SOD2 were highly upregulated in mesenchymal GBM (Figure 1F). Subsequently, we determined the expression of DEGs in mesenchymal GBM cells, GBM tumors with high Nrf2 expression, and GBM tumors. We examined the DEGs of each crucial phenotype and determined that 610 genes were shared between those phenotypes; the results are depicted in a Venn diagram (Figure 1G). Among other oxidoreductase-related genes, *CYBB* and *SOD2* were shared and highly expressed in GBM tumors, mesenchymal GBM cells, and tumors with high NRF2 expression (Figure 1H). At both the single-cell and bulk levels, consistent aberrant expression of *CYBB* and *SOD2* indicated high Nrf2 activation, and both *CYBB* and *SOD2* were differentially expressed in mesenchymal GBM cells and linked to treatment resistance.

### 2.2. CYBB Characterized the Poor Outcomes of Mesenchymal GBM Patients

To validate the presence of aberrant CYBB expression in the clinical setting, we examined the tumor specimens of 65 patients with GBM from Taipei Medical University-Shuang Ho Hospital (TMU–SHH). Immunostaining analysis indicated that the tumor specimens exhibited relatively high CYBB expression along with the overexpression of other well-known mesenchymal markers (Figure 2A). A comparative analysis demonstrated significantly higher expression levels of N-cadherin, CD44, and vimentin in patients with GBM with high *CYBB* expression, suggesting the remarkable expression of *CYBB* in mesenchymal GBM cells (Figure 2B).

Moreover, in the TCGA–GBM dataset, we observed that high *CYBB* expression was significantly associated with poor progression-free survival (PFS) in patients with GBM (Figure 2C). The results of the Kaplan–Meier subgroup analysis by subtype indicated a significant correlation between *CYBB* levels and lower PFS in mesenchymal GBM only (Figure 2D). However, Cox regression analysis demonstrated that higher *CYBB* expression was an independent risk factor for significantly poorer progression-free survival in patients with GBM after the adjustment for subtypes (hazard risk = 1.6; 95% confidence interval = 1.05–2.4; *p* = 0.029) (Figure 2E). The findings indicate that *CYBB* expression affected the clinical outcomes of patients with GBM, suggesting the aggressive features of mesenchymal GBM.

### 2.3. Resilience to Ferroptosis Is Co-Associated with TMZ Resistance in Mesenchymal GBM

To determine the specific molecular alterations conferring TMZ resistance in mesenchymal GBM cells, we determined the differential transcriptomic profiles of most GBM cell lines. We performed a heatmap cluster analysis of most GBM cell lines to identify the mesenchymal subtype. The U87MG cell line was selected because it exhibited high activation of mesenchymal signatures along with high Nrf2 signaling, invasiveness, and resistance gene set scores (Figure 3A). To examine TMZ resistance in an in vitro model, we used an approach employed in a previous study with some modifications and generated TMZ-resistant clones of U87MG-R cells with an IC50 higher than that of parental cells (Figure 3B). In addition, U87MG-R cells proliferated more rapidly than parental U87MG (Appendix A), confirming that resistance to TMZ treatment is unlikely due to the slower growth rate of U87MG-R than parental GBM cells. Western blotting results revealed that TMZ-resistant U87MG-R cells highly expressed CYBB along with Nrf2, SOD2, and other epithelial–mesenchymal transition (EMT) markers, such as slug, vimentin, N-Cadherin, and CD44, in addition to the upregulation of cancer stemness marker CD133 (Figure 3C). In addition, TMZ-resistant U87MG-R cells exhibited a significantly increased number of tumor spheres (Figure 3D) and an aggressiveness phenotype in terms of invasion and migration (Figure 3E). Therefore, TMZ resistance in mesenchymal GBM was associated with aberrant *CYBB* expression, Nrf2/SOD2 mitochondrial antioxidant axis activation, and cancer stemness properties.

To evaluate the resilience of TMZ-resistant mesenchymal GBM to ferroptosis, we treated U87MG-R cells with erastin to induce ferroptosis. Here, TMZ-resistant cells exhibited higher mitochondrial oxidative stress along with higher mitochondrial mass, as indicated by the higher fluorescence intensity of MitoSOX and MitoTracker in U87MG-R cells than in U87MG cells (Figure 3F). Moreover, TMZ-resistant U87MG-R cells exhibited higher resilience against erastin than parental U87MG cells did (Figure 3G). The PI staining assay was performed to determine the number of dead cells following ferroptosis induction (erastin and TBHP) with or without cotreatment with ferroptosis inhibitors (SRS1192 and NAC). We observed a greater reduction in the ferroptotic cell death percentage in U87MG-R cells than in parental cells following the induction of ferroptosis with or without cotreatment with a ferroptosis inhibitor, suggesting the attenuation of ferroptosis-specific cell death sensitivity in TMZ-resistant mesenchymal GBM cells (Figure 3H). These findings indicated that TMZ-resistant GBM cells exhibited higher resilience to ferroptosis induction with a greater mitochondrial oxidative stress and higher mitochondrial mass. Thus, on the basis of the finding of the activation of the Nrf2/SOD2 mitochondrial antioxidant axis, certain interactions between CYBB and that axis might help in understanding the susceptibility of TMZ-resistant GBM cells to ferroptosis.

### 2.4. Molecular Interaction between CYBB and Nrf2 Promoted TMZ Resistance in GBM

To comprehend the generalizability of CYBB in contributing to the development of TMZ resistance across GBM cells, several TMZ-resistant GBM cell lines with different mesenchymal properties were generated and tested. According to gene set enrichment analysis across multiple GBM cell lines (Figure 3A), cell lines other than U87MG were then selected as representative of other mesenchymal (Hs683) and non-mesenchymal (T98G) GBM cell lines. The protein level of CYBB and mesenchymal markers expression were then compared between U87MG, Hs683, and T98G cells (Appendix A). Both U87MG and Hs683 cells displayed relatively higher mesenchymal markers such as CD44, Vimentin, and N-cadherin along with CYBB than T98G, suggesting high mesenchymal activation in both cells, whereas a lack of mesenchymal marker upregulation in T98G cells denoted the non-mesenchymal properties of this cell. Next, Hs683 and T98G GBM cell lines were serially exposed to TMZ, similar to the previous step in generating U87MG-R cells. Both Hs683-R and T98G-R exhibited higher resistance to TMZ treatment than their respective parental cell lines (Appendix A). Silencing of CYBB via short-hairpin RNA (shRNA)-mediated knockdown re-sensitized both Hs683-R and T98G-R cells upon TMZ treatment (Appendix A). Moreover, knockdown of CYBB repressed Nrf2 and SOD2 expression while deactivating mesenchymal markers such as CD44, Slug, Vimentin, and N-Cadherin (Appendix A). Interestingly, even in the non-mesenchymal-derived parental GBM cells such as T98G, silencing of CYBB in the TMZ-resistant clones of T98G-R still showed a potential effect of repressing NRF2 and SOD2 upregulation while downregulating mesenchymal markers to potentiate tumor suppression of TMZ treatment (Appendix A). Therefore, the data confirmed the general contribution of CYBB in supporting TMZ resistance irrespective of the original properties of the mesenchymal phenotype in GBM cells, despite the level of impact potentially varying between mesenchymal and non-mesenchymal GBM cells.

To determine the functional role of CYBB in regulating the mesenchymal features and treatment resistance of GBM, we performed shRNA-mediated knockdown of *CYBB*. U87MG-R cells with *CYBB* suppression exhibited fewer mesenchymal features, including decreased numbers of projected and slender cells, than scramble control cells did (Figure 4A). Moreover, the drug–response curve indicated that the knockdown of *CYBB* increased the sensitivity of U87MG-R cells against TMZ treatment, suggesting that *CYBB* plays a role in increasing TMZ resistance (Figure 4B). In addition, the knockdown of *CYBB* downregulated Nrf2 and SOD2 expression while deactivating the expression of several mesenchymal markers, such as CD44, Slug, and Vimentin, in TMZ-resistant U87MG-R cells (Figure 4C). In terms of cancer stemness characteristics, the depletion of *CYBB* reduced the invasion and migration capacities of U87MG-R cells and the number of tumor spheres (Figure 4D,E), suggesting the crucial role of CYBB in mediating the EMT and self-renewal of TMZ-resistant cells. Immunofluorescence staining revealed a decline in Nrf2 expression following *CYBB* suppression in U87MG-R cells, indicating the interdependence of Nrf2 expression upon *CYBB* suppression in TMZ-resistant cells (Figure 4F). To explore protein–protein interactions between Nrf2 and CYBB, we performed a coimmunoprecipitation assay. CYBB was physically bound to Nrf2, as indicated by a high expression of Nrf2 following the formation of immune complex precipitation by CYBB (Figure 4G). The findings indicate that the molecular interaction of CYBB with Nrf2 led to a shift in the mesenchymal phenotype, cancer stemness promotion, and resistance development in GBM.

### 2.5. Nrf2/SOD2 Axis Abrogated the Oxidative Stress and Ferroptosis of TMZ-Resistant GBM

Nrf2, a redox-sensitive transcription factor, governs the expression of genes involved in endogenous antioxidant synthesis. Considering the presence of higher mitochondrial oxidative stress in TMZ-resistant mesenchymal GBM cells, we examined the putative role of the mitochondrial antioxidative capacity under the activation of the Nrf2/SOD2 circuitry. The promoter site of SOD2 was examined using the JASPAR dataset, and the specific site of Nrf2 motif binding was identified (Figure 5A). We performed a luciferase assay of the SOD2 promoter of the Nrf2 binding site in U87MG-R cells to investigate the transcriptional activity of SOD2. Treatment with both oxidative stress inducers, TBHP and TMZ, increased the transcription of SOD2. Inhibition of NOX through treatment with GSK2795039 reduced SOD2 transcription (Figure 5B). This finding indicated that Nrf2-mediated SOD2 transcription was a compensatory mechanism in response to the induction of high oxidative stress by extrinsic treatment with drugs, such as TMZ, or intrinsically generated ROS from NOX.

We knocked down *SOD2* by using short-hairpin RNA in TMZ-resistant GBM cells to understand its functional role (Figure 5C). Both erastin and TMZ treatment have previously been found to accumulate ROS and trigger ferroptosis in several GBM cell lines, including U87MG cells [12,13]. Here, a more detailed role of SOD2 in regulating mitochondrial ROS level in chemoresistant GBM U87MG-R cells was further examined. U87MG-R cells with *SOD2* suppression exhibited a higher level of mitochondrial oxidative stress following treatment with TBHP, TMZ, and erastin (Figure 5D). This finding suggested the crucial role of SOD2 in attenuating high oxidative stress in TMZ-resistant GBM cells. Knockdown of SOD2 sensitized TMZ-resistant GBM cells to erastin-mediated ferroptosis, as indicated by a depression in the drug–response curve (Figure 5E), marked suppression of the colony-forming ability (Figure 5F), and upregulation of the ferroptosis marker Prostaglandin Synthase-2 (*PTGS2*) (Figure 5G). In addition, the ferroptotic cell death percentage was higher in U87MG-R cells with SOD2 suppression than in scramble control cells following TBHP and erastin treatment (Figure 5H). These findings indicated that the Nr2/SOD2 axis was mainly involved in the abrogation of high oxidative stress and ferroptosis induction in TMZ-resistant mesenchymal GBM cells.

### 2.6. Erastin Induced Ferroptosis in TMZ-Resistant GBM Cells with SOD2 Suppression In Vivo

To determine the preclinical efficacy of an erastin analog, imidazole ketone erastin (IKE), in overcoming TMZ resistance in GBM, we performed an in vivo study in which mice were xenografted with TMZ-resistant U87MG-R cells. Moreover, we examined the antiferroptotic role of SOD2 by comparing the efficacy of this analog between scramble and *SOD2*-suppressed tumor xenografts. Treatment with IKE significantly suppressed the growth of scramble TMZ-resistant GBM xenografts. Increased tumor growth inhibition was observed in *SOD2*-suppressed mice xenografts treated with IKE (Figure 6A). This finding indicated the possibility of enhancing tumor suppression following erastin treatment by targeting *SOD2* to overcome TMZ resistance in GBM. In addition, no difference in mouse body weight was observed between mice subjected to *SOD2* suppression or IKE treatment and those subjected to vehicle and scramble control treatment (Figure 6B). This finding indicated the general tolerability of IKE treatment in the in vivo setting. Moreover, the immunostaining of xenograft tumors revealed that the knockdown of *SOD2* promoted the activity of the lipid peroxidation marker 4HNE and the oxidative stress accumulation marker 8-OHDG as well as enhanced the induction of erastin-mediated ferroptosis, as indicated by the increased expression of the ferroptosis marker PTGS2 (Figure 6C,D). Therefore, erastin-mediated ferroptosis could effectively overcome TMZ resistance in GBM cells in vivo. Exploiting *SOD2* might optimize the overall efficacy of erastin.

## 3. Discussion

Adult glioblastomas are the most prevalent and aggressive types of primary brain tumors. Despite the availability of intensive combinatorial chemotherapy, the survival rate for GBM remains low, with a median overall survival of approximately 1 year [1]. Treatment-refractory and recurrent GBM is largely caused by heterogeneity within and between tumors. Because of their highly aggressive nature, GBM tumors harboring the mesenchymal signature have drawn considerable interest [3]. EGFR and PDGFRA amplifications are prevalent in proneural and classical GBMs, respectively. The NF1 deficiency occurs in mesenchymal GBMs mainly through homozygous and hemizygous deletions. Loss of NF1 results in the infiltration of tumor-associated macrophages into the tumor microenvironment, causing mesenchymal transition and radioresistance to chemotherapy [2,3,18]. In addition to suggesting a unique microenvironment orchestration, our study results suggest that mesenchymal GBMs contain certain deregulated oxidoreductases. We identified a NOX subunit, called CYBB, that can help in the detection but can also determine mesenchymal signature activation in GBM cells.

Glioblastoma with the mesenchymal subtype is highly invasive and expresses mesenchymal and proinflammatory genes. Cancer cells must retain mesenchymal and stemness properties to survive because stem-cell-like cells are resistant to treatment and promote tumor development. Our results revealed that CYBB expression was significantly associated with Nrf2 activation, CD44 overexpression, and N-cadherin overexpression in mesenchymal GBM cells. Nrf2 activation is commonly observed during mesenchymal GBM transformation [19]. Thus, Nrf2 activation may serve as a surrogate marker for the prognosis of mesenchymal GBM. Furthermore, in this study, we noted that CYBB expression was associated with the aggressiveness of GBM. In gliomas exhibiting Nrf2 overactivity, CYBB might control the expression of antioxidant enzymes involved in drug detoxification and alleviate redox stress. Therefore, targeting events downstream of CYBB may provide an opportunity for clinicians to exploit this specific subpopulation of GBM to provide precision medicine.

The mesenchymal subtype of GBM is more aggressive and resistant to multiple therapies compared with other subtypes, implying the presence of metabolic abnormalities, such as altered redox homeostasis, due to an elevated metabolic rate and a harsh tumor microenvironment [5]. NOX, a major source of intracellular superoxide, plays a significant role in regulating cellular redox signaling and homeostasis. NOX2, a member of the NOX family that was first identified to be expressed in myeloid cells, has been extensively studied in both normal and cancerous cells [8]. The membrane-bound subunits CYBB (also called NOX2 or gp91phox, where phox refers to phagocyte oxidase) and p22phox (CYBA) are responsible for the catalytic core of oxidase. CYBB activation can result in the accumulation of ROS and contribute to the development of a resistant phenotype in acute myeloid leukemia (AML). AML with a high expression level of CYBB is associated with poor prognosis and chemoresistance [20]. In this study, the aberrant expression of *CYBB* and the antioxidant gene *SOD2* resulted in Nrf2 activation, and both these genes were differentially expressed in mesenchymal GBM. Therefore, understanding the interaction among CYBB, NRF2, and SOD2, which were markedly active in mesenchymal GBM, can explain the orchestration of pro-oxidative and antioxidative regulation that underlies treatment resistance in mesenchymal GBM.

Although ROS mediate key intracellular processes, such as carcinogenesis and tumor proliferation, their targeting causes collateral damage to host cells [8]. ROS-mediated activation of Nrf2 is a crucial negative regulation step that can mitigate the detrimental effects of excessive ROS levels because NRF2 stimulates the expression of antioxidant enzymes. Neutrophil Cytosolic Factor 1 (NCF1) or p47-phox is physically bound to Nrf2, preventing the ubiquitination and activation of Nrf2 [21]. However, our findings indicated that CYBB might physically interact with Nrf2 to promote its subsequent upregulation, thus causing mesenchymal shift, cancer stemness, and TMZ resistance in GBM. Cancer cells utilize Nrf2 transcriptional networks to counteract oxidative stress. Furthermore, tumor cells produce more antioxidative enzymes than normal cells, presumably to compensate for the toxicity of ROS generated by NOX [8]. Because of the dual effects of ROS on both resistance and cell death, this study emphasizes that CYBB-mediated oxidative stress supports the mesenchymal features of GBM, activates Nrf2 to protect GBM cells from high ROS levels, and enhances the resistance of GBM cells to TMZ.

TMZ affects the bioenergetics and dynamics of mitochondria. Considerable changes in mitochondrial DNA and electron transport chain remodeling occur in response to TMZ-induced stress in GBM cells. The mitochondrial enzyme SOD2 was identified as a target of specificity protein 1 (Sp1); the increase in this protein in resistant cells is another mechanism underlying chemoresistance independent of methylguanine methyltransferase (MGMT) [22]. The findings of this study indicated that Nrf2 activation upon aberrant CYBB expression regulates the transcription of SOD2 to promote TMZ resistance. This mechanism is vital to abrogate mitochondrial superoxide generation after exposure to TMZ. Other disease models have revealed that Nrf2 induces mitochondrial antioxidant enzymes, such as Sirt3 and SOD2, to maintain mitochondrial ROS homeostasis and protect neurons from oxidative damage [23]. In addition, Nrf2 activation stimulates peroxisome proliferator-activated receptor coactivator 1α expression to protect cells against oxidative stress damage and enhance cisplatin resistance in ovarian cancer cells [24]. Recent studies have identified SOD2 as a specific mitochondrial antioxidant system associated with TMZ resistance in mesenchymal GBM. Thus, the identification of alternative pro-oxidant approaches may help in overcoming acquired TMZ resistance.

Dixon et al. investigated how erastin selectively kills RAS-mutated tumor cells. Their research led to the discovery of ferroptosis, a cell death mechanism that considerably differs from apoptosis, necrosis, and autophagy in terms of cellular morphology, biochemistry features, and the key genes involved [11]. Our results indicated that erastin-induced ferroptosis may kill TMZ-resistant GBM cells despite the mitochondrial antioxidant circuitry activity of Nrf2/SOD2. In response to TMZ treatment or ferroptosis inducers, such as erastin and TBHP, the suppression of *SOD2* expression resulted in the accumulation of mitochondrial superoxide. The superoxide radical is generated when an oxygen molecule receives an electron. By contrast, superoxide dismutase (SOD) is crucial for cellular defense against superoxide. Mitochondrial SOD2 converts superoxide into peroxide, which diffuses into the cytosol. Thus, SOD is essential to attenuate ferroptosis induction by inhibiting radical chain reactions.

The results of this study corroborate the previous reports that have highlighted the importance of SOD2 in buffering superoxide accumulation, as well as dictating sensitivity to chemotherapy and pro-ferroptosis agents [7,25,26]. Specifically, our study showed that the depletion of SOD2 in TMZ-resistant GBM cells re-sensitized them to erastin-mediated ferroptosis, both in vitro and in vivo. Aside from that, inhibition of SOD2 increased mitochondrial ROS upon erastin or TMZ treatment. It has been known that the major form of ROS produced within mitochondria is superoxide [8]. Previous studies have shown that decreased cellular antioxidant capacity, such as SOD2 levels, contributes to ferroptosis [26,27]. The antioxidative function of SOD2 is located within the mitochondrial matrix, where it converts superoxide anions produced by mitochondria during electron transport chain (ETC) to a less harmful radical, hydrogen peroxide (H_2_O_2_). As a result of the conversion of hydrogen peroxide to water by catalase, peroxidases, or peroxiredoxins, the cell is further preserved from deleterious oxidative stress. On the other hand, SOD2 deficiency leads to superoxide accumulation in the mitochondria, leading to oxidative stress. As both erastin and TMZ treatment produce ROS in cells [13,26], a depletion of SOD2 in the mitochondria can further increase superoxide levels. A higher superoxide level leads to increased lipid peroxidation in membranes and the occurrence of ferroptosis [28]. In essence, mitochondrial oxidative stress resulting from SOD2 depletion can facilitate ferroptosis and increase the sensitivity of glioma cells to erastin or TMZ.

The susceptibility of TMZ-resistant GBM cells to ferroptosis may be explained by the interaction between CYBB and Nrf2/SOD2. In response to extrinsic treatment with drugs, such as TMZ, or intrinsically created ROS from NOX, Nrf2-mediated SOD2 transcription is a compensatory response. This study demonstrated that Nrf2/SOD2 prevented ferroptosis activation and excessive ROS production. Ferroptosis may be promoted by blocking Nrf2 or its downstream antioxidant genes through cotreatment with inducers, such as erastin and sorafenib [29]. The suppression of *SOD2* can markedly increase ferroptosis sensitivity and suppress TMZ resistance in mesenchymal GBM. The findings of our animal study revealed that erastin-induced ferroptosis was highly effective in overcoming TMZ resistance. Targeting SOD2 may increase its efficacy in GBM. Thus, high CYBB expression may indicate not only mesenchymal features but also ferroptosis susceptibility in TMZ-resistant GBM. Erastin treatment may benefit mesenchymal GBM by exerting pro-oxidant effects and overcoming TMZ resistance. Because an erastin analog has not yet been tested in clinical trials, the use of tolerable drugs that inhibit SOD2 may promote ferroptosis while eliminating TMZ resistance in mesenchymal GBM; however, future studies on this topic are warranted.

## 4. Materials and Methods

### 4.1. Reanalysis of Single-Cell RNA Sequencing Dataset

We selected a representative single-cell transcriptome profiling dataset reported by Yu et al.; they used this dataset to explore molecular cross-talks between heterogeneous glioma cells and the microenvironment. We reanalyzed this dataset to determine the expression of the genes of interest in each cell cluster [30]. The dataset by Yu et al. is deposited in the Genome Sequence Archive (GSA) database (accession number, HRA000179). In the GSA database, this dataset is named as the Chinese Glioma Genome Atlas (CGGA) and consists of approximately 8000 cells. After obtaining the file matrix, we used the Seurat package (version 4.0.6) in the R environment (version 4.0.1) to reconstruct the Seurat object. General preprocessing steps were performed, including the filtering of genes with no expression and reducing noise from the weakly expressed mitochondrial genome. Subsequently, the normalization and data scaling of the Seurat object were performed, followed by reducing the dimension and generation of cell clusters by employing the t-Distributed Stochastic Neighbor Embedding (t-SNE) module. The positive and negative markers of each cluster were generated and listed. The t-SNE plot, dot plot, and bar graph were constructed to present differences in the expression of the genes of interest between clusters.

### 4.2. Pathway Enrichment Analysis

To investigate key signaling pathways and biological processes perturbed in specific cell clusters, the Seurat module score was employed. The module score function was used to calculate the feature expression of each cell on the basis of selected gene set signatures or annotation. We used several functional annotations and curated gene sets following the previous gene signature to identify the subtype of GBM [2]. Subsequently, we displayed the module score in an array of the t-SNE plot.

### 4.3. TCGA–GBM Dataset Acquisition

Bulk tumor RNA-sequencing data from TCGA for 172 patients with GBM, namely TCGA–GBM, were downloaded from the Xenabrowser portal, including their clinical information. The RNA-sequencing dataset is normalized and transformed by the web portal and presented in the “log2” format. The overall survival data of each patient were collected for survival analysis. 

### 4.4. Cell Line Transcriptome Profiling

The Cancer Cell Line Encyclopedia dataset, which contains the RNA-sequencing profiles of almost all commercial glioma cell lines, was downloaded from the DepMap portal [31]. Gene enrichment analysis of mesenchymal signatures and other functional annotations was performed using gene set variation analysis. A heatmap was generated to present the enrichment level of gene signatures in each cell line.

### 4.5. Identification of Differentially Expressed Genes

A normalized microarray dataset was prepared, and phenotypes for each sample were preset. The DESeq2 package (version 3.52.2) was used to count the fold-change difference and significance level for each gene. Genes with a |log2 fold-Change| of ≥1 and a *p* value of <0.05 were considered as differentially expressed genes (DEGs). DEGs from each dataset were presented in a Venn diagram to determine shared and common DEGs between the phenotypes of interest. 

### 4.6. Immunohistochemical Tissue Staining

We detected the expression levels of CYBB protein and mesenchymal markers, namely vimentin (Vim), CD44, and N-cadherin, in 65 GBM tissue specimens from the TMU–SHH GBM cohort by performing immunohistochemical (IHC) analysis. Formalin-fixed, paraffin-embedded tissue sections were placed on coated glass slides. IHC analysis was performed in accordance with the standard protocol involving antigen retrieval, primary antibody incubation, secondary chromogen addition, and hematoxylin counterstaining. Rabbit anti-CYBB monoclonal antibody was used at a dilution of 1:200. Images were captured using a microscope (Leica Microsystems, Wetzlar, Germany). Immunostaining expression of each tissue specimen was scored using the IHC Q or quick score on the basis of the intensity and extent of expression. The Q-score method has been described previously. The intensity was scored as follows: 0 = negative, 1 = weak, 2 = medium, and 3 = high. The extent of expression was evaluated as the percentage of the positively stained area compared with the entire tumor area. The final IHC Q score was determined by quantifying the intensity score and positivity percentage. The minimum and maximum IHC scores were 0 and 300, respectively. The tissue specimens were categorized into high and low CYBB expression groups by using the median Q score of the entire cohort as the cutoff. The association of the IHC Q score with certain clinical parameters of each patient was evaluated.

### 4.7. Cell Line Culture

The human GBM cell line U87MG was purchased from the American Type Culture Collection (Manassas, VA, USA). U87MG cells primarily possess the mesenchymal phenotype [32]. The cells were cultured in Dulbecco’s modified Eagle’s medium (Invitrogen Life Technologies, Carlsbad, CA, USA) supplemented with 10% fetal bovine serum (FBS) and 1% penicillin/streptomycin (Invitrogen). The cells were grown at 37 °C in an incubator with 5% humidified CO_2_. Cell passage was performed when the cells had almost reached full confluence. The culture medium was replaced with fresh medium every 72 h before the administration of TMZ or any other treatment.

### 4.8. Generation of TMZ-Resistant Cells

TMZ-resistant U87MGR cells were generated by treating them at increasing doses (up to 150 µM) of TMZ for 2 weeks, followed by maintenance treatment with 100 µM TMZ in vitro in accordance with a previous study [33]. Chemosensitivity was evaluated by performing the sulforhodamine B viability assay, and the corresponding drug–response curve was plotted to indicate the generation of TMZ resistance in U87MGR cells.

### 4.9. shRNA-Mediated Knockdown

The lentiviral CYBB shRNA (Cat#: TL313616V) and SOD2 shRNA (Cat#: TL309190V) were obtained from Origene (Rockville, MD, USA) and used to suppress the expression of the corresponding genes in accordance with the manufacturer’s protocol. The nontarget sequence (Scr shRNA, Origene Cat#: TR30021V) was used as control. Transfection was performed for 24 h by using Lipofectamine 2000 in accordance with the manufacturer’s instructions. The knockdown efficiency was verified using immunoblotting or quantitative polymerase chain reaction.

### 4.10. Immunofluorescence Staining

Immunofluorescence staining was performed to determine changes in particular markers under specific conditions in GBM cells. Prior to staining, either scramble or shCYBB U87MG-R cells were cultured in six-well plates and treated with 100 µM TMZ for 24 h or left untreated. Subsequently, the cells were plated in six-well chamber slides and permeabilized by treating them with 0.1% Triton X-100 in 0.01 M phosphate-buffered saline (PBS, pH 7.4). The cells were washed twice in PBS containing 1% bovine serum albumin and stained with rabbit anti-Nrf2 (dilution 1:100, cat. #7631, Cell Signaling, Danvers, MA, USA) overnight. The stained cells were washed three times, resuspended in a mounting medium, and fixated onto coverslips. Then, 4′,6-diamidino-2-phenylindole (DAPI) was used for nuclear staining. Photographs were obtained using a Leica spectral confocal fluorescence imaging system.

### 4.11. Cell Death Analysis

Each resistant or parental cell line, with either CYBB or SOD2 knockdown, was exposed to either TMZ, erastin, or tert-butyl hydroperoxide (TBHP) with or without cotreatment with ferroptosis inhibitors for approximately 24 h. Subsequently, the cells were harvested, washed with PBS, and fixed with ice-cold 70% ethanol at −20 °C for 30 min. Then, cells were incubated with 5 µg/mL RNase for 30 min at room temperature and stained with propidium iodide (PI, 5 µg/mL) for 1 h. Dead cells were stained with PI and examined through flow cytometry.

### 4.12. Tumorsphere Study

The cells were seeded in serum-free low-adhesion culture plates containing RPMI1640 with B27 supplement (Invitrogen), 20 ng/mL EGF, and 20 ng/mL basic FGF (stem cell medium; PeproTech, Rocky Hill, NJ, USA) for approximately 14 days to allow the formation of tumorspheres. Spheres were counted under a microscope. The tumorsphere formation efficiency was calculated as the ratio of the number of tumorspheres formed to the seeded adherent cell number.

### 4.13. Colony Formation Study

To observe the generation of tumor colonies, approximately 2500 GBM cells/cm^2^ were suspended in 0.3% agarose containing the tumorsphere medium (StemCell Technologies, Vancouver, BC, Canada) and layered on a 0.8% agar base layer. The culture was covered with 0.5 mL of tumorsphere medium and cultured for 14 days. For quantification, the wells were observed under a microscope, and tumor colonies were analyzed using ImageJ software.

### 4.14. Invasion and Migration Study

To evaluate the migration and invasion capabilities of cancer cells, Transwell migration and invasion assays were performed. Briefly, U87MG or U87MG-R cells were incubated in six-well plates for 24 h until they reached full confluency. Matrigel precoating was required for invasion studies before seeding with 1 × 10^5^ cells, whereas migration studies did not require precoating the membrane. In two chambers, different concentrations of BSA were added, with the upper chamber containing RPMI with 2% and the lower chamber containing 20% FBS. Incubation was conducted for 48 h, either with or without treatment. Cells that penetrated membranes were carefully removed, and those that migrated or invaded were fixed in ethanol and stained with crystal violet. Under a microscope, five randomly selected fields were examined, and representative images were captured to determine the number of migrated or invaded cells.

### 4.15. Mitochondrial ROS Study

The superoxide indicator MitoSOX red (Invitrogen, Karlsruhe, Germany) was used to examine mitochondrial ROS production. MitoTracker Green FM (Invitrogen, Karlsruhe, Germany) was used to detect mitochondria. Prior to staining, cells were grown in six-well plates and treated with 100 µM TMZ, 150 µM TBHP, 10 µM Erastin, or 2 µM SRS11-92 for 12 h or left untreated. The cells were then detached, the medium was replaced, and approximately 30,000 cells/well were regrown on eight-well slides. MitoSOX (1 µM) and MitoTracker Green FM (150 nM) were applied for 20 min to U87MG and U87MG-R cells. Hoechst dye was used to counterstain the cells. The cells were fixed with 4% paraformaldehyde for 20 min at room temperature. A fluorescence microscope was used to obtain images.

### 4.16. Reporter Luciferase Assay

The pGL3-SOD2 promoter Luc reporter gene construct was generated using a plasmid pGL3-basic (Promega, Madison, WI, USA) containing the luciferase gene and a human SOD2 promoter fragment as described previously [34,35] to evaluate the transcriptional activity of the SOD2 promoter. The SOD2 promoter region spanning from −3340 to +260 nucleotides was cloned into a pGL3 basic vector. The putative Nrf2 or antioxidant response element binding site was located at +65 to +76 of the cloned SOD2 promoter. The reporter gene construct was transfected into U87MG-R cells by using Lipofectamine 2000 in accordance with the manufacturer’s protocol. Eighteen hours after the transfection, the cells were exposed to TBHP or TMZ or GSK2795039 (NOX inhibitor) for twenty-four hours. Luciferase activity was determined using a dual-luciferase reporter assay kit (Promega, Madison, WI, USA).

### 4.17. Tumor Xenograft Animal Study

Animal experiments were performed using 5-to-6-week-old female BALB/C nude mice obtained from BioLASCO (Taipei City, Taiwan). Animals were maintained in an animal center at 22–28 °C under a 12 h diurnal cycle and provided unrestricted access to food and water. Using Matrigel mixture (Becton Dickinson, Bedford, MA, USA), we subcutaneously injected 100 µL solution containing 1 × 10^6^ scrambled or shSOD2 U87MG-R cells into the left flank of each mouse as part of in vivo studies (The IACUC Approval Number LAC-2021-0578). Imidazole ketone erastin (IKE) was used as an erastin analog to induce ferroptosis in mice harboring either scramble or shSOD2 U87MG-R xenografts. A total of four groups were formed by randomly dividing the mice: control group consisting of shScramble xenografts treated with vehicle (n = 5), shScramble xenografts treated with IKE (n = 5), shSOD2 xenografts treated with vehicle (n = 5), and shSOD2 xenografts treated with IKE (n = 5). Drug treatment was started after the tumor volume reached approximately 100 mm^3^ or 10 days after the injection of the tumor xenograft. Depending on the treatment group, vehicles (5 mg/kg/day) or IKE (25 mg/kg/day) were administered intraperitoneally. Treatment was administered intraperitoneally for 3 weeks, and tumor growth was observed for 6 weeks after treatment began. Tumor growth was evaluated every 2 weeks by using Vernier calipers, and tumor volume (V) was calculated using the following formula: V = 0.5 × [long diameter × short diameter^2^]. Tumor mass was collected in week 6 after mice were sacrificed and used in further experiments.

### 4.18. Statistical Analysis

Numerical variables are presented as the mean and standard deviation, whereas categorical variables are expressed as the frequency and percentage. An association between categorical variables was assessed using the chi-square test. Student’s *t*-test was used to compare continuous variables within two groups, and one-way analysis of variance (ANOVA) was used to evaluate discrepancies among more than two groups. Two-way ANOVA was used to determine the differential responses of numerical variables by two independent variables. Kaplan–Meier analysis was performed to estimate survival curves, and the log-rank test was performed to calculate *p* values. Cox regression was performed using univariate and multivariate methods to assess the effect of the risk group on the survival of patients with GBM. Statistical significance was set at *p* < 0.05. Data analysis was performed using GraphPad Prism 6.0 (San Diego, CA, USA) and SPSS version 21.0 (IBM, Armonk, NY, USA).

## 5. Conclusions

In this study, as shown in the schematic illustration of Figure 7 we demonstrated the crucial role of CYBB in promoting the acquisition of TMZ resistance in GBM, resulting in mesenchymal transformation. The disruption of the Nrf2/SOD2 antioxidant circuitry resulted in the vulnerability of mesenchymal GBM cells to ferroptosis, potentially overcoming TMZ resistance.

## Figures and Tables

**Figure 1 ijms-24-07706-f001:**
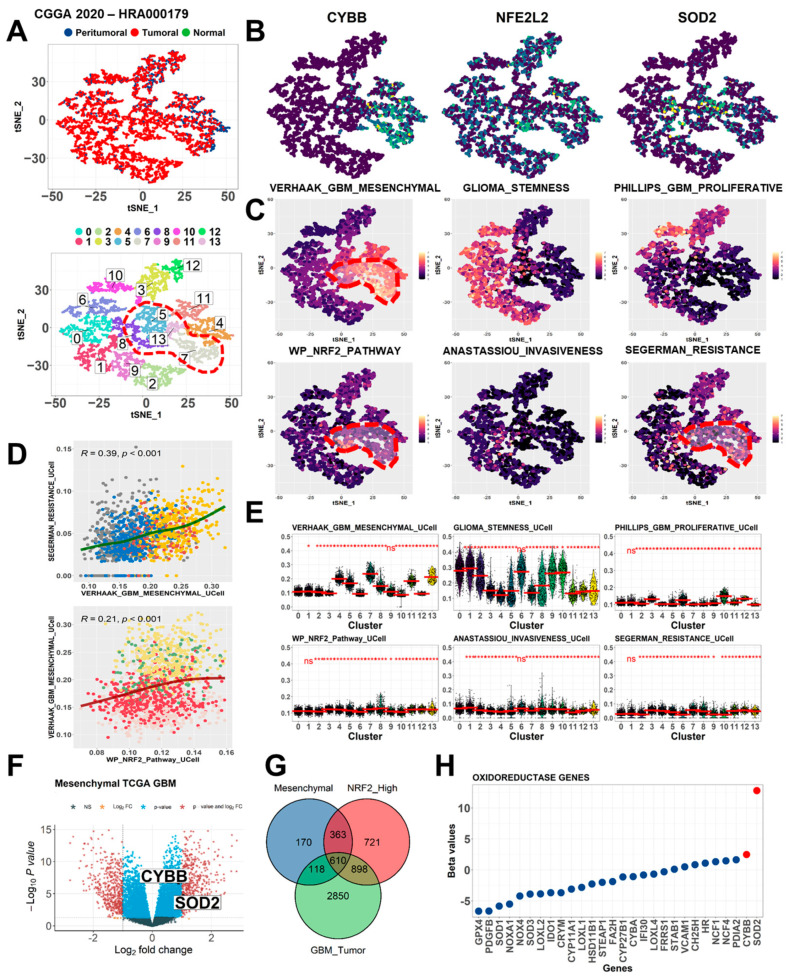
Cluster of glioma tumor with aberrant CYBB exhibited mesenchymal features, temozolomide (TMZ) resistance, and Nr2 pathway signature activation. (**A**) Representative t-Distributed Stochastic Neighbor Embedding (t-SNE) plots of the Chinese Glioma Genome Atlas (CGGA) dataset show cluster cells grouped by the sample type (top panel) and number of clusters (bottom panel). Putative cell clusters with high CYBB expression were contained in an arbitrary border (dotted red line). (**B**) Array of t-SNE plots shows the expression levels of the putative main targets of this study (CYBB, NFE2L2 (or Nrf2), and SOD2). (**C**) t-SNE plots depict the module score, which represents the degree of activation of each gene set signature. Module scores of clusters with aberrant CYBB expression were delineated in an arbitrary border (dotted red line). (**D**) Scatter plot illustrates the co-association of each gene set signature of interest in terms of their module scores. Pearson’s coefficient and *p* value are provided in the top margin. (**E**) Violin plots present the comparison of module scores in each cell cluster. The median module score is labeled with a red line in each violin bar. Cluster number 0 served as control due to the presence of predominant tumoral cells with low CYBB expression. (**F**) Volcano plot depicts the overexpression of CYBB and SOD2 among DEGs between mesenchymal and nonmesenchymal GBM tumor cells in the TCGA-GBM dataset. (**G**) Venn diagram represents shared common DEGs among mesenchymal GBM, GBM with high NRF2 expression, and GBM tumor; CYBB and SOD2 are commonly expressed genes. (**H**) Dot plot depicts the significance of oxidoreductase genes in mesenchymal GBM. Significant genes are presented as red dots (SOD2 and CYBB), while the blue dots represent the genes below minimum threshold of log fold change. Significance level: ns: not significant, * *p* < 0.05.

**Figure 2 ijms-24-07706-f002:**
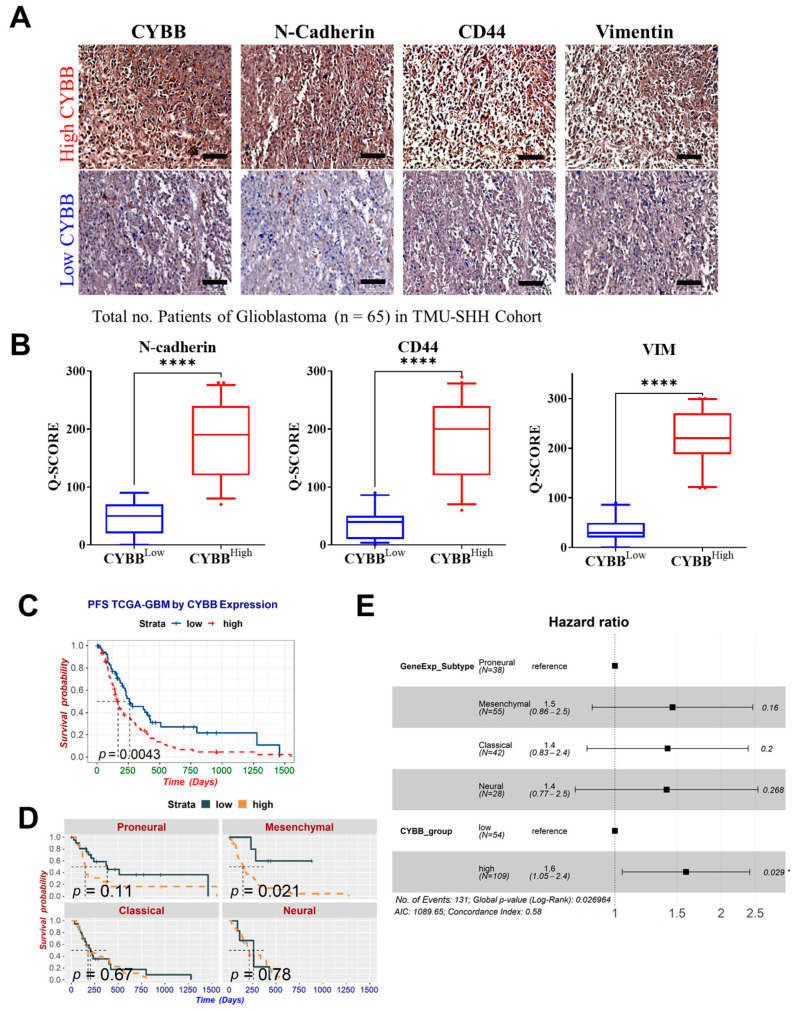
Aberrant CYBB expression was associated with mesenchymal glioblastoma multiforme (GBM) and poor outcomes. (**A**) Representative images of the staining of mesenchymal markers in the tissue specimen of the TMU-SHH GBM cohort by CYBB expression grouping, which is classified according to the median immunohistochemistry (IHC) Q score as the cutoff. (**B**) Violin plot compares the IHC Q scores of GBM with high and low CYBB expression and presents the differential expression of mesenchymal markers. (**C**) Kaplan–Meier curve presents the association between CYBB expression and progression-free survival (PFS) of patients with GBM in the TCGA-GBM dataset. (**D**) Subgroup analysis of Kaplan–Meier curves indicated the presence of differential PFS between high and low CYBB expression groups based on annotated subtypes. (**E**) Forest plot depicts the adjusted hazard ratio of CYBB expression and GBM subtypes determined by performing multivariate Cox regression analysis in accordance with PFS of GBM cohort in TCGA dataset. Significance level: * *p* < 0.005, **** *p* < 0.0001. Scale bar: 200 µm.

**Figure 3 ijms-24-07706-f003:**
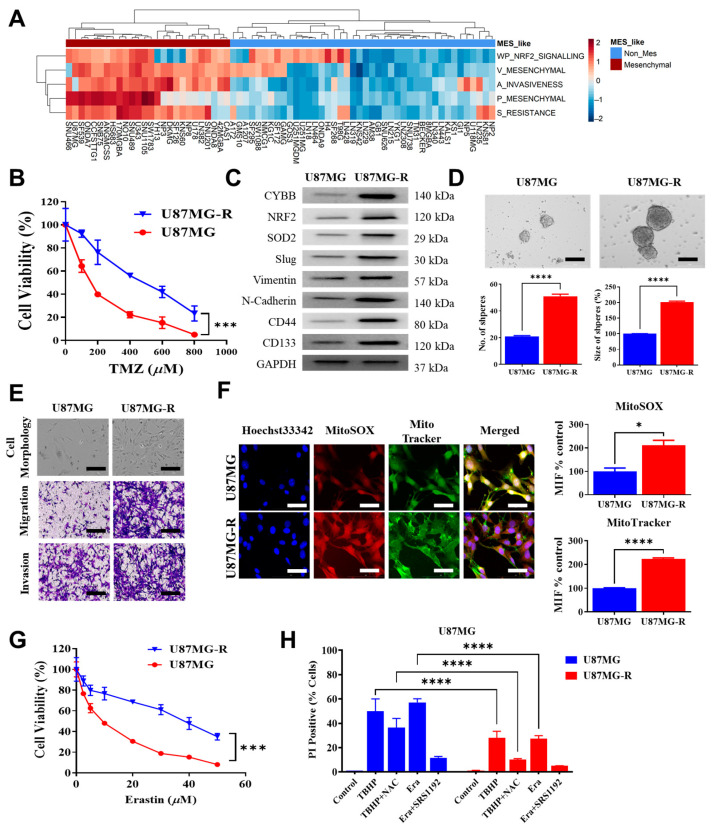
Temozolomide (TMZ) resistance was associated with mesenchymal features, ferroptosis resilience, and elevated mitochondrial ROS state. (**A**) Heatmap plots present the enrichment score of functional signatures in each cell line. Cell lines with putative mesenchymal-like enrichment are marked by a red box in the top panel, and cell lines without mesenchymal features are indicated by a blue box. (**B**) The drug–response curve describes differential TMZ sensitivity between parental U87MG and resistant U87MG-R glioma cell lines. (**C**) Western blotting indicates the upregulation of CYBB, Nrf2, SOD2, mesenchymal markers (slug, vimentin, N-cadherin, CD44), and stemness marker (CD133) in TMZ-resistant U87MG-R. (**D**) Representative images of the tumor sphere assay and corresponding quantification are provided. Quantification bar displays the differential extent of tumor sphere generation in terms of the number and size of spheres between parental and resistant cells. (**E**) Representative images of the basic morphology of cell lines and migration and invasion assays are presented. Mesenchymal-like U87MG-R cells exhibited a spindle shape. Populations of migrated and invaded cells were intensified in U87MG-R. (**F**) Immunofluorescence images illustrate the comparative expression of the mitochondrial ROS marker MitoSOX and the mitochondrial label MitoTracker between U87MG and U87MG-R cells. Bar chart quantification of the mean fluorescence intensity of the corresponding marker and cell lines is presented. (**G**) Drug–response curve represents different sensitivity to erastin in U87MG and U87MG-R. (**H**) Bar chart compares the percentage of cell death, as shown by cells stained with propidium iodide following the induction of ferroptosis by treatment with either 150 µM tert-butyl hydroperoxide (TBHP) or 10 µM erastin with or without ferroptosis inhibitors (5 mM N-acetyl cysteine (NAC) or 2 µM SRS11-92) between U87MG and U87MG-R. Significance level: * *p* < 0.05; *** *p* < 0.001; **** *p* < 0.0001. Scale bar in D & E: 100 µm, F: 50 µm.

**Figure 4 ijms-24-07706-f004:**
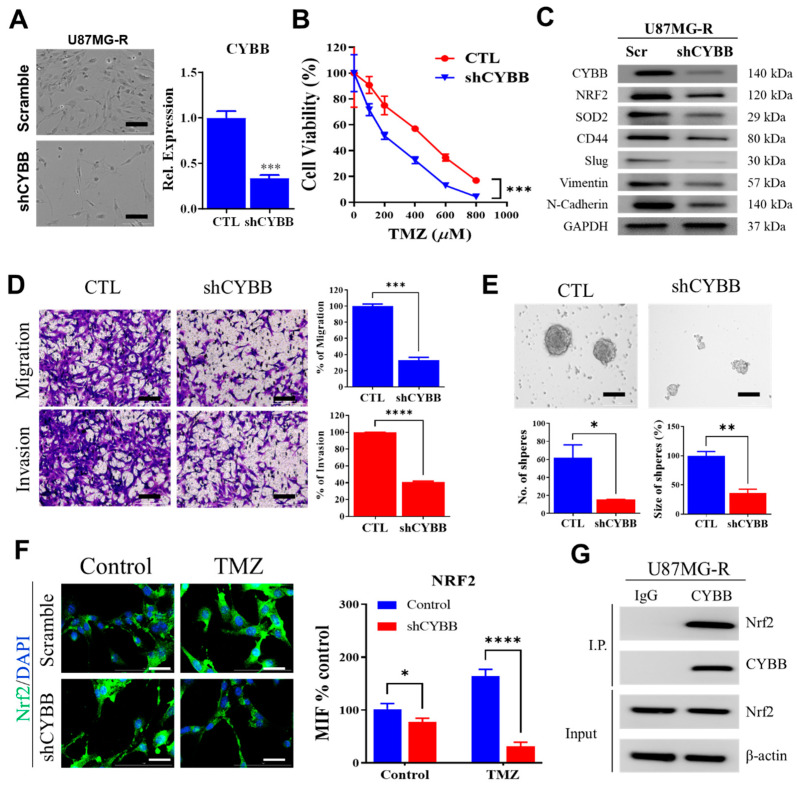
Binding of CYBB to Nrf2 promotes mesenchymal features and temozolomide (TMZ) resistance. (**A**) Representative images of cellular morphology are presented following shRNA-mediated knockdown of CYBB (shCYBB) in U87MG-R cells. Spindle-shaped mesenchymal-like cells were less evident in shCYBB cells. Knockdown efficiency was observed through real-time quantitative reverse transcription polymerase chain reaction and is presented with a quantification bar chart. (**B**) Drug–response curve describes different sensitivity to TMZ treatment between scramble and shCYBB U87MG-R cells. (**C**) Western blotting indicated the downregulation of Nrf2, SOD2, and mesenchymal markers (CD44, Slug, Vimentin, and N-Cadherin) upon repression of CYBB. (**D**) Representative images of migration and invasion study between scramble and shCYBB U87MG-R cells are presented. Population of migrated and invaded cells are presented in a bar chart. (**E**) Images of tumor sphere study in scramble and shCYBB U87MG-R cells are presented. Generation of tumor spheres in terms of the number and size of spheres was quantified in a corresponding bar chart. (**F**) Immunofluorescence images illustrate the comparative expression of Nrf2 between scramble and shCYBB U87MG-R cells. Bar chart quantification of mean fluorescence intensity is presented. (**G**) U87MG-R cell lysate was incubated with the CYBB/p91-phox antibody or isotypic IgG. The immune complex precipitation revealed the coprecipitation of Nrf2 with CYBB/p91-phox (top panel). CYBB/p91-phox precipitation occurred and was identified using the corresponding antibody (second panel). The total cell lysate as input was immunoblotted for Nrf2 (third panel) and β-actin (bottom panel). Significance level: * *p* < 0.05; ** *p* < 0.01; *** *p* < 0.001; **** *p* < 0.0001. Scale bar in A, D & E: 100 µm, F: 20 µm.

**Figure 5 ijms-24-07706-f005:**
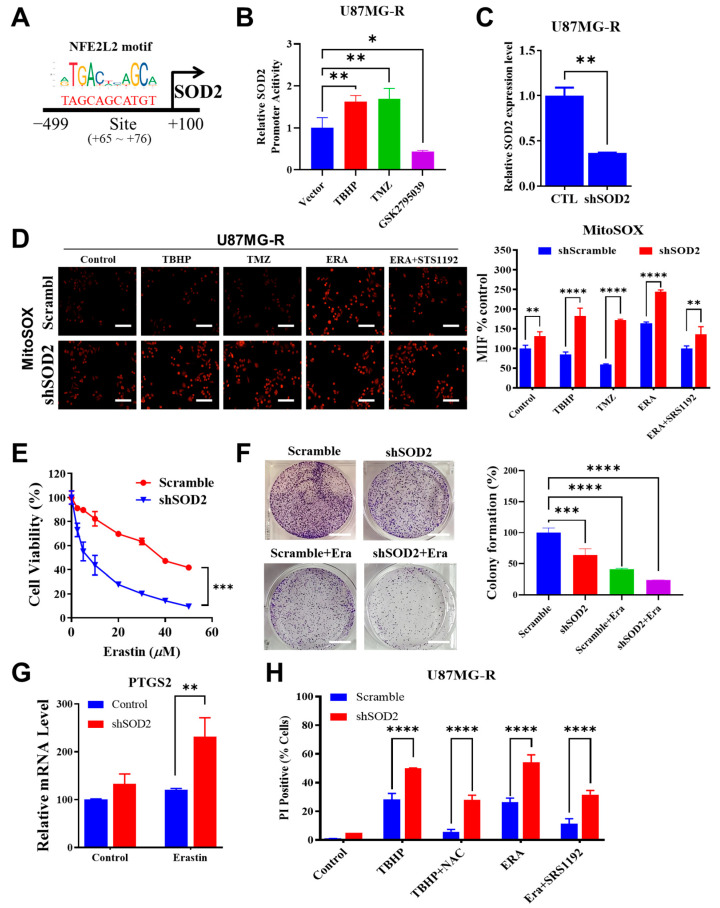
Nrf2-mediated SOD2 transcription abrogates mitochondrial reactive oxygen species (ROS) and ferroptosis. (**A**) Schematic depicts the predicted binding site of NFE2L2 (Nrf2) in the SOD2 promoter by JASPAR. (**B**) Bar graph shows SOD2 promoter reporter activity in U87MG-R transfectants. U87MG-R cells were transiently transfected with a plasmid containing the SOD2 luciferase reporter under the pGL3-SOD2 or with plasmid pRL-TK as a transfection control. Cells were treated with tert-butyl hydroperoxide (TBHP), temozolomide (TMZ), or GSK2795039 (NOX inhibitor) for 24 h. (**C**) Bar chart illustrates SOD2 knockdown efficiency in U87MG-R cells through qRT-PCR. (**D**) Immunofluorescence images illustrate the comparative expression of mitochondrial ROS generation, as indicated by MitoSOX expression, between scramble and shSOD2 in U87MG-R cells. Bar chart of mean fluorescence intensity is presented. (**E**) Drug–response curve presents different sensitivity to erastin between scramble and shSOD2 of U87MG-R cells. (**F**) Representative result of colony formation following erastin treatment and SOD2 knockdown is presented. The quantification bar chart of colony percentages is shown. (**G**) Bar chart depicts the modulation of the ferroptosis marker PTGS2 upon SOD2 knockdown and erastin treatment in U87MG-R cells through qRT-PCR. (**H**) Bar chart shows the percentage of cell death, as indicated by cells stained with propidium iodide following the induction of ferroptosis by treatment with either 150 µM TBHP or 10 µM erastin with or without ferroptosis inhibitors (5 µM N-acetyl cysteine or 2 µM SRS11-92) between U87MG and U87MG-R. Significance level: * *p* < 0.05; ** *p* < 0.01; *** *p* < 0.001; **** *p* < 0.0001. Scale bar in D: 100 µm, F: 400 µm.

**Figure 6 ijms-24-07706-f006:**
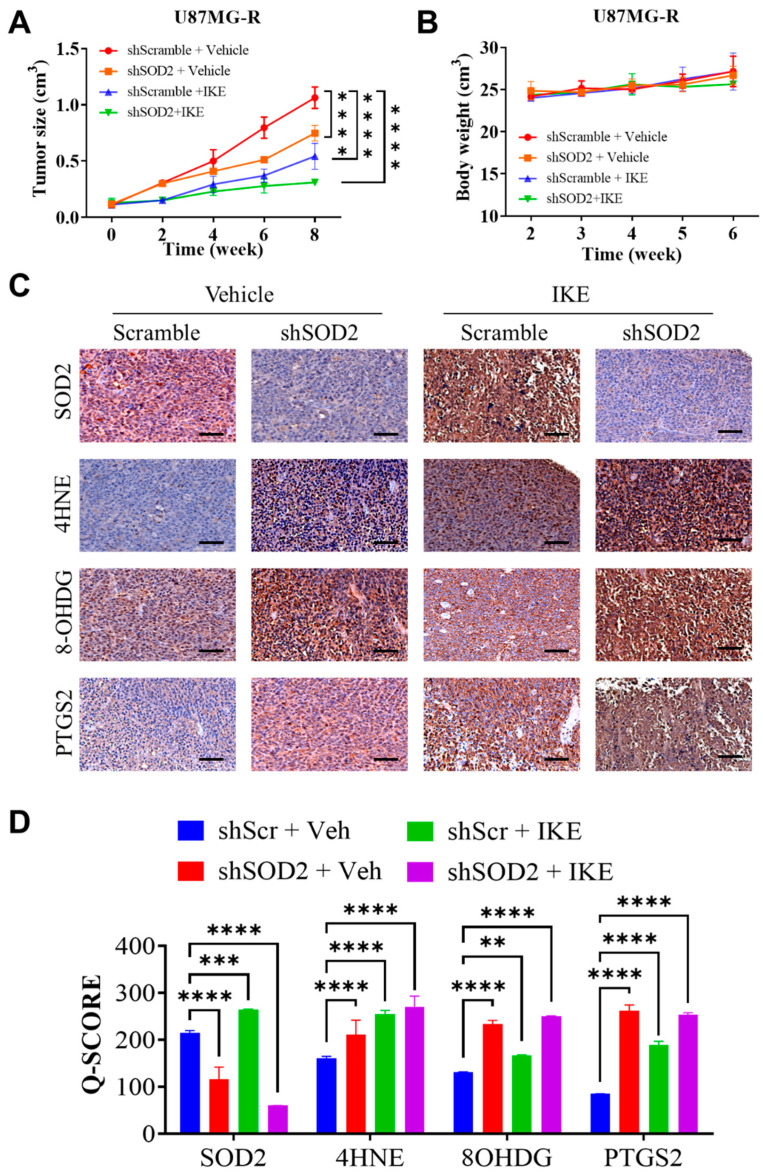
Erastin analog induced ferroptosis in a SOD2-dependent manner in vivo. (**A**) Line graph depicts the differential trend of tumor size in each mice group following treatment with an erastin analog and SOD2 repression. (**B**) The line chart presents no changes in body weight in each mice group. (**C**) Representative figures of immunostaining indicate the modulation of several markers, such as SOD2, the lipid peroxidation marker 4HNE, the oxidative stress marker 8-OHDG, and the ferroptosis marker PTGS2, following treatment. (**D**) Bar graph depicts the quantitative modulation of each marker expression, represented by the immunohistochemistry Q score, following treatment. Significance level: ** *p* < 0.01; *** *p* < 0.001; **** *p* < 0.0001. Scale bar: 200 µm.

**Figure 7 ijms-24-07706-f007:**
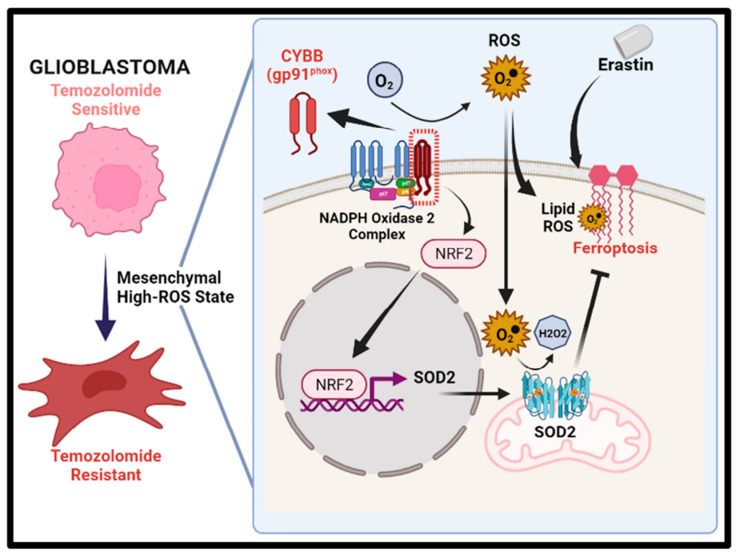
Schematic illustration of how CYBB contributes to temozolomide resistance and ferroptosis resilience in mesenchymal glioblastoma. The left panel indicates that temozolomide resistance is associated with high oxidative stress and mesenchymal shift. The right panel illustrates that aberrant CYBB expression induced ROS accumulation and activated Nrf2/SOD2 circuitry, thus perturbing ferroptosis resilience upon erastin treatment.

## Data Availability

The datasets used and analyzed in the current study are publicly accessible as indicated in the manuscript.

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
