# Peer review of "NADPH Oxidase Subunit CYBB Confers Chemotherapy and Ferroptosis Resistance in Mesenchymal Glioblastoma via Nrf2/SOD2 Modulation"

_ijms, 2023, doi:10.3390/ijms24097706_

Round 1

Reviewer 1 Report

This manuscript examines the role of CYBB in glioblastoma progression and resistance to treatment specifically focusing on the mesenchymal sub-group. It is a good manuscript, however some points need to be addressed. These include: 

1. Does TMZ induce ROS production?

2. It is concerning when only one cell line is used. The authors should consider using at least another mesenchymal glioblastoma cell line

3.Does TMZ induce ferroptosis in U87 parental cells in context to U87R cells.

4. To determine if the U87 system is a good clinically significant model, the authors should show if CYBB expression in the U87 pair of cells (Fig 3C) correlates with N-Cadherin, CD44 and Vimentin as shown in the clinical  samples (Fig 2B).

5. Likewise, does knockdown of CYBB reverse these effects on N-Cadherin, CD44 and Vimentin (Fig 4C).

6. The cell viability assay (Fig 3B and 4B) masks whether the cells grow at the same rate as both are normalised to 100%. Do the U87R cells grow slower than the parentals and if so, is the resistance to TMZ simply due to having less effect on slower growing cells or cells with less dividing potential?

Author Response

Dear Reviewer,

Coauthors and I very much appreciated the encouraging, critical and constructive comments on this manuscript by the reviewer. The comments have been very thorough and useful in improving the manuscript. We strongly believe that the comments and suggestions have increased the scientific value of the revised manuscript by many folds. We have taken them fully into account in revision. We are submitting the corrected manuscript with the suggestion incorporated in the manuscript. The manuscript has been revised as per the comments given by the reviewer, and our responses to all the comments are as follows:

Response to Reviewers:

Reviewer #1: This manuscript examines the role of CYBB in glioblastoma progression and resistance to treatment specifically focusing on the mesenchymal sub-group. It is a good manuscript, however some points need to be addressed. These include:

  1. Does TMZ induce ROS production?

Answer: It was a pleasure to receive the reviewer's thoughtful comment. TMZ has been known to induce ROS generation in glioma cell lines according to the findings of several studies previously. Because those findings have been reported in multiple reports, we then consider not replicating the potentially similar result and redundant data to our study. However, the findings of previous studies relevant to this question will be provided in the introduction section. Therefore, please kindly refer to our Introduction.

Updated Introduction Section, please see page 2, line 85-97.

Ferroptosis is a form of regulated cell death governed by excess iron and charac-terized by ROS accumulation and lipid peroxidation, which cause membrane rupture. This nonapoptotic form of cell death was first described in 2012 by a study employing erastin to induce ferroptosis [11]. Ferroptosis is effective in eliminating treatment-resistant tumor cells. Moreover, ferroptosis has been linked with drug resistance upon TMZ treatment. TMZ treatment has been known to accumulate ROS level either the mitochondrial or lipid ROS, increase labile iron pool, while depleting anti-oxidative capacity in glioma cells [12, 13]; thereby indicating ferroptosis induction as alternative TMZ mechanism of action in glioblastoma. On the other hand, resistance upon TMZ also closely connects with the way how glioma cells defend upon deleterious im-pact of high ROS state and ferroptosis resilience [7, 14]. Establishing this link may ena-ble the use of ferroptosis to treat gliomas. Nrf2 inhibition increased the sensitivity of glioma cells to TMZ [15]. In addition, TMZ treatment promoted and generated gluta-thione to confer resistance [16].

Updated Reference Section, please page 20, line 782-785.

  1. Su, J.; Li, Y.; Liu, Q.; Peng, G.; Qin, C.; Li, Y., Identification of SSBP1 as a ferroptosis-related biomarker of glioblastoma based on a novel mitochondria-related gene risk model and in vitro experiments. Journal of Translational Medicine 2022, 20, (1), 440.
  2. Song, Q.; Peng, S.; Sun, Z.; Heng, X.; Zhu, X., Temozolomide Drives Ferroptosis via a DMT1-Dependent Pathway in Glioblastoma Cells. Yonsei Med J 2021, 62, (9), 843-849.

  1. It is concerning when only one cell line is used. The authors should consider using at least another mesenchymal glioblastoma cell line

Answer: We thank the editor for their critical suggestion. Indeed additional cell line may confirm and increase generalizability of the finding. According to this comment, we then provided other mesenchymal and non-mesenchymal glioblastoma cell line along with the key results of CYBB/Nrf2/SOD2 axis contribution to determine TMZ resistance in our supplementary result. Therefore, please kindly refer to our Result section

Updated Result Section, please see page 8, line 280-303.

To comprehend generalizability of CYBB in contributing development of TMZ resistance across GBM cells, several TMZ-resistant GBM cell lines with different mesenchymal properties were then generated and tested. According to gene set enrichment analysis across multiple GBM cell lines (Fig. 3A), cell lines other than U87MG were then selected as representative of other mesenchymal (Hs683) and non-mesenchymal (T98G) GBM cell lines. The protein level of CYBB and mesenchymal markers expression were then compared between U87MG, Hs683, and T98G cells (Fig. S2A). Both U87MG and Hs683 cells displayed relatively higher mesenchymal markers such as CD44, Vimentin, and N-cadherin along with CYBB than T98G, suggesting high mesenchymal activation in both cells, whereas lacking of mesenchymal marker upregulation in T98G cells denoted non-mesenchymal properties of this cell. Next, Hs683 and T98G GBM cell lines were serially exposed to TMZ similar to the previous step in generating U87MG-R cells. Both Hs683-R and T98G-R exhibited resistance to TMZ treatment than their respective parental cell lines (Fig. S2B and S2C). In resonance to the finding in U87MG-R cells (Fig. 4B), silencing of CYBB re-sensitized both Hs683-R and T98G-R cells upon TMZ treatment (Fig. S2B and S2C). Moreover, knockdown of CYBB repressed Nrf2 and SOD2 expression while deactivating mesenchymal markers such CD44, Slug, Vimentin, and N-Cadherin (Fig. S2D). Interestingly, even in the non-mesenchymal-derived parental GBM cells such as T98G, silencing of CYBB in the TMZ-resistant clones of T98G-R still showed potential effect of repressing NRF2 and SOD2 upregulation while downregulating mesenchymal markers to potentiate tumor suppression of TMZ treatment (Fig. S2C and S2D). Therefore, the data confirmed the general contribution of CYBB in supporting TMZ resistance irrespective to the original properties of mesenchymal phenotype in GBM cells, despite the level of impact might vary between mesenchymal and non-mesenchymal GBM cells.

Updated figure S2, please see supplementary section.

Updated caption figure S2, please see supplementary section.

Figure S2. Role of CYBB in mesenchymal and non-mesenchymal GBM cell lines in the acquisition of TMZ resistance. (A) Western blotting indicated differential expression of CYBB and mesenchymal activation according markers (CD44, Vimentin, N-Cadherin) between presumed mesenchymal cells (U87MG, Hs683) and non-mesenchymal (T98G) GBM cells. The drug response curve showed relative differential sensitivity of TMZ treatment upon CYBB silencing in two chemoresistant GBM cell lines: Hs683 (B) and T98G (C). (D) Western blotting portrayed perturbation of NRF2/SOD2 axis activation and mesenchymal markers (CD44, Slug, Vimentin, N-Cadherin) between presumed Hs683-R and T98G-R chemoresistant GBM cells.

  1. Does TMZ induce ferroptosis in U87 parental cells in context to U87R cells.

Answer: We appreciate the editor for their interesting question. There have been several report that already demonstrated TMZ treatment might indeed ferroptosis in U87 parental cells even though with different underlying pathway modulation. For example, Su et al. showed that TMZ exposure reduced viability of U87MG while increasing mitochondrial ROS, reduced FTH1 expression while accumulating labile iron and depleting GSH pool which indeed characterized ferroptosis induction. As we would like to determine mitochondrial ROS generation upon TMZ and observe any induction ferroptosis may occur in chemoresistant GBM cells, we consider to not reduplicating similar finding in U87 parental cells as previous study has already demonstrated quite similar result. This finding summary of previous studies regarding induction of ferroptosis by TMZ would be provided and delivered in our introduction and result section. Please kindly refer to our introduction and result section.

Updated Introduction Section, please see page 2, line 85-97.

Ferroptosis is a form of regulated cell death governed by excess iron and charac-terized by ROS accumulation and lipid peroxidation, which cause membrane rupture. This nonapoptotic form of cell death was first described in 2012 by a study employing erastin to induce ferroptosis [11]. Ferroptosis is effective in eliminating treatment-resistant tumor cells. Moreover, ferroptosis has been linked with drug resistance upon TMZ treatment. TMZ treatment has been known to accumulate ROS level either the mitochondrial or lipid ROS, increase labile iron pool, while depleting anti-oxidative capacity in glioma cells [12, 13]; thereby indicating ferroptosis induction as alternative TMZ mechanism of action in glioblastoma. On the other hand, resistance upon TMZ also closely connects with the way how glioma cells defend upon deleterious im-pact of high ROS state and ferroptosis resilience [7, 14]. Establishing this link may ena-ble the use of ferroptosis to treat gliomas. Nrf2 inhibition increased the sensitivity of glioma cells to TMZ [15]. In addition, TMZ treatment promoted and generated gluta-thione to confer resistance [16].

Updated Result Section, please see page 11,  line 359-364.

Either erastin or TMZ treatment has been studied previously to accumulate ROS and trigger ferroptosis in several GBM cell lines, including U87MG cells [12, 13]. Here more detailed role of SOD2 in regulating mitochondrial ROS level in chemoresistant GBM U87MG-R cells was further examined. U87MG-R cells with SOD2 suppression exhibit-ed a higher level of mitochondrial oxidative stress following treatment with TBHP, TMZ, and erastin (Fig. 5D).

Updated Reference Section, please page 20, line 782-785.

  1. Su, J.; Li, Y.; Liu, Q.; Peng, G.; Qin, C.; Li, Y., Identification of SSBP1 as a ferroptosis-related biomarker of glioblastoma based on a novel mitochondria-related gene risk model and in vitro experiments. Journal of Translational Medicine 2022, 20, (1), 440.
  2. Song, Q.; Peng, S.; Sun, Z.; Heng, X.; Zhu, X., Temozolomide Drives Ferroptosis via a DMT1-Dependent Pathway in Glioblastoma Cells. Yonsei Med J 2021, 62, (9), 843-849.

  1. To determine if the U87 system is a good clinically significant model, the authors should show if CYBB expression in the U87 pair of cells (Fig 3C) correlates with N-Cadherin, CD44 and Vimentin as shown in the clinical samples (Fig 2B).

Answer: We thank the editor for their constructive advice. Indeed, by determining N-cadherin, CD44, and vimentin, this might convince mesenchymal activation in U87MG GBM cells. As per the suggestion, we then added western blotting result pertaining to several mesenchymal markers (N-cadherin, CD44, and vimentin) in our result. Therefore, please kindly refer to our result.

Updated Result Section, please see figure 3C in page 7, line 236.

Updated caption figure 3C, please see page 7, line 242-244.

Figure 3. (C) Western blotting indicates the upregulation of CYBB, Nrf2, SOD2, mesenchymal markers (slug, vimentin, N-cadherin, CD44), and stemness marker (CD133) in TMZ-resistant U87MG-R.

Updated Result Section, please see page 6, line 210-213.

Western blotting results revealed that TMZ-resistant U87MG-R cells highly expressed CYBB along with Nrf2, SOD2, and other epithelial–mesenchymal transition (EMT) markers, such as slug, vimentin, N-Cadherin, and CD44 in addition to upregulation of cancer stemness marker CD133 (Fig. 3C).

  1. Likewise, does knockdown of CYBB reverse these effects on N-Cadherin, CD44 and Vimentin (Fig 4C).

Answer: We thank the editor for their constructive advice. Indeed, by determining N-cadherin, CD44, and vimentin, expression in relation to CYBB silencing this might convince contribution of CYBB in activating mesenchymal transition of U87MG GBM cells. As per the suggestion, we then added western blotting result pertaining to several mesenchymal markers (N-cadherin, CD44, and vimentin) in our result. Therefore, please kindly refer to our result.

Updated Result Section, please see figure 4C in page 9, line 305.

Updated caption figure 4C, please see page 9, line 312-313.

Figure 4. (C) Western blotting indicated the downregulation of Nrf2, SOD2, and mesenchymal markers (CD44, Slug, Vimentin, N-Cadherin) upon repression of CYBB.

Updated Result Section, please see page 8, line 265-268.

In addition, the knockdown of CYBB downregulated Nrf2 and SOD2 expression while deactivating expression of several mesenchymal markers such as CD44, Slug, and Vi-mentin in TMZ-resistant U87MG-R cells (Fig. 4C).

  1. The cell viability assay (Fig 3B and 4B) masks whether the cells grow at the same rate as both are normalised to 100%. Do the U87R cells grow slower than the parentals and if so, is the resistance to TMZ simply due to having less effect on slower growing cells or cells with less dividing potential?

Answer: We appreciate the editor for such critical and positive comments. According to this comment we then determine tumor proliferation assay between U87MG-R and parental U87MG and resulted accelerated growth rate in chemoresistant U87MG-R. This result indicated that the resistance to TMZ is unlikely contributed by the cells which grow in a slower rate. Therefore, please kindly refer to our Result and Supplementary section.

Updated figure S1, please see supplementary section.

Updated caption figure S1, please see supplementary section.

Figure S1. Increased cellular proliferation rate in U87MG-R than parental U87MG cells. Significance level: *** p < 0.001.

Updated Result Section, please see page 6, line 205-210.

To examine TMZ resistance in an in vitro model, we used an approach employed in a previous study with some modifications and generated TMZ-resistant clones of U87MG-R cells with a IC50 higher than that of parental cells (Fig. 3B). In addition, U87MG-R cells proliferated more rapidly than parental U87MG (Fig. S1), confirming that resistance to TMZ treatment is unlikely due to the slower growth rate of U87MG-R than parental GBM cells.

Reviewer 2 Report

In the submitted manuscript I-Chang Su and colleagues describe how activity of CYBB/Nrf2/SOD2 proteins affects the viability and invasiveness of glioblastoma cells upon genotoxic insult and ferroptosis induction. The investigated proteins respond to and regulate the reactive oxygen species levels within the different compartments of glioblastoma cells, exerting the significant effect on cell phenotype and drug resistance. The observations provided in the manuscript are also clinically relevant. The experimental methods are appropriate, and experimental evidence provided is solid. The majority of experiments was done in cells resistant to temozolomide (TMZ) toxicity, and this approach is substantiated by the primary/acquired temozolomide resistance observed in glioblastoma patients. Actually, due to the activation of CYBB/Nrf2/SOD2 axis, TMZ-resistant cells are concomitantly resistant to ferroptosis. 

MAJOR POINTS:

1.       The correlation of CYBB overexpression/overactivation and mesenchymal features of GBM, is shown in patient transcriptomic data set, independent IHC staining set, and U87MG cells treated with TMZ. CYBB protects GBM cells against TMZ toxicity. However, the particular mesenchymal markers, like vimentin, slug etc are not studied in CYBB-silenced U87MG-R cells, and this experiment would justify the claim made that CYBB actually triggers mesenchymal phenotype in GBM cells.

2.       Second part of the story tackling CYBB/Nrf2/SOD2 axis activation is more complete. However, the high levels of ROS produced by mitochondria of cells resistant to TMZ and thus displaying high levels of SOD2 antioxidant activity need to be further discussed.

3.       The way how statistical significance is presented on the graphs is not satisfactory. Very often it is not clear which bars/ data points are actually compared. For example, four bars are compared with four bars in figure 3G, which is not correct and not explanatory. Figure 4F – which difference is statistically significant? The same with data quantification in Figure 5B, 5D, 5F, 6D.

4.       The figure quality is sometimes low (Figure 1), thus figure text is impossible to decipher. Another issue is straight copy-pasting of the graphs from Excel (?) without further editing, therefore fonts are often disfigured and too small.

5.       All experiments were done with use of the one and single GBM cell line. The comparison of the results obtained in U87MG and other cell lines would let to determine how general is the effect. Have Authors made any attempt to check CYBB increase and its consequences upon TMZ treatment in other cell lines? Also, U87MG is described as “mesenchymal” and it gets even more mesenchymal when treated with TMZ. Therefore, performing experiments on non-mesenchymal cell line would be more appropriate and let to study the mesenchymal shift.

1.       The manuscript title is somehow incomprehensible/misleading. Actually, the provided data show clearly that CYBB promotes both chemoresistance (due to the mesenchymal shift?) and resistance to ferroptosis due to SOD2 activation (not vulnerability to ferroptosis). 

MINOR POINTS:

1.       Figure 4C – X-axis description is incorrect, blue and red bars indicate scramble and shCYBB conditions, respectively.

2.       Figure 3 is loaded with plenty of information; to make it more comprehensible it could be split into two figures: one figure on CYBB-TMZ-mesenchymal story and second figure on CYBB-Nrf2-SOD2-erastin story. Alternatively, moving to 3F and 3G subpanels to the end of the figure, before mito ROS measurements (subpanel) needs to be considered.

3.       Figure 2C, 2D – what is actually shown in the Kaplan-Meyer curve graphs: overall survival or progression free survival?

4.       Whenever abbreviation appears in the text, the full name should be provided (e.g. for CYBB, Nrf2, PTGS2, TMU-SHH and so on).

5.       Line 91: “promoted the generation of glutathione to confer the resistance” instead of “promoted and generated glutathione…”

6.       Line 92: sorafenib is primarily an RTK inhibitor, not ferroptosis inducer

7.       Line 299: “… cells originating from tumor components were more than those from peritumoral or…” what actually means that “cells were more”? Please, reformulate this sentence.

Author Response

Reviewer #2: In the submitted manuscript I-Chang Su and colleagues describe how activity of CYBB/Nrf2/SOD2 proteins affects the viability and invasiveness of glioblastoma cells upon genotoxic insult and ferroptosis induction. The investigated proteins respond to and regulate the reactive oxygen species levels within the different compartments of glioblastoma cells, exerting the significant effect on cell phenotype and drug resistance. The observations provided in the manuscript are also clinically relevant. The experimental methods are appropriate, and experimental evidence provided is solid. The majority of experiments was done in cells resistant to temozolomide (TMZ) toxicity, and this approach is substantiated by the primary/acquired temozolomide resistance observed in glioblastoma patients. Actually, due to the activation of CYBB/Nrf2/SOD2 axis, TMZ-resistant cells are concomitantly resistant to ferroptosis.

MAJOR POINTS:

  1. The correlation of CYBB overexpression/overactivation and mesenchymal features of GBM, is shown in patient transcriptomic data set, independent IHC staining set, and U87MG cells treated with TMZ. CYBB protects GBM cells against TMZ toxicity. However, the particular mesenchymal markers, like vimentin, slug etc are not studied in CYBB-silenced U87MG-R cells, and this experiment would justify the claim made that CYBB actually triggers mesenchymal phenotype in GBM cells.

Answer: We thank the editor for their constructive advice. This comment is actually also asked by other reviewer that mention mesenchymal markers should be investigated in shCYBB U87MG-R cells. As per the suggestion, we then added western blotting result pertaining to several mesenchymal markers (CD44, Slug, vimentin, and N-cadherin) in our result. Therefore, please kindly refer to our result.

Updated Result Section, please see figure 4C in page 9, line 305.

Updated caption figure 4C, please see page 9, line 312-313.

Figure 4. (C) Western blotting indicated the downregulation of Nrf2, SOD2, and mesenchymal markers (CD44, Slug, Vimentin, N-Cadherin) upon repression of CYBB.

Updated Result Section, please see page 8, line 265-268.

In addition, the knockdown of CYBB downregulated Nrf2 and SOD2 expression while deactivating expression of several mesenchymal markers such as CD44, Slug, and Vi-mentin in TMZ-resistant U87MG-R cells (Fig. 4C).

  1. Second part of the story tackling CYBB/Nrf2/SOD2 axis activation is more complete. However, the high levels of ROS produced by mitochondria of cells resistant to TMZ and thus displaying high levels of SOD2 antioxidant activity need to be further discussed.

Answer: We thank the editor for this critical yet positive comment. According to this comment, the more relevant explanation of SOD2 contribution as an important antioxidant in GBM to increase TMZ resistance and ferroptosis resilience will be discussed more in our Discussion section. Therefore, please kindly refer to our Discussion section.

Updated Discussion Section, please see page 14, line 490-508.

The results of this study support previous studies that have highlighted the importance of SOD2 in buffering superoxide accumulation, as well as dictating sensitivi-ty to chemotherapy and pro-ferroptosis agents [7, 31, 32]. Specifically, our study showed that depletion of SOD2 in TMZ-resistant GBM cells re-sensitized to erastin-mediated ferroptosis, both in vitro and in vivo. Aside from that, inhibition of SOD2 in-creased mitochondrial ROS upon erastin or TMZ. It has been known that the major form of ROS produced within mitochondria is superoxide [8]. Previous studies have shown that decreased cellular antioxidant capacity, such as SOD2 levels, contributes to ferroptosis [32, 33]. The anti-oxidative function of SOD2 is located within the mitochondrial matrix, where it converts superoxide anions produced by mitochondria during electron transport chain (ETC) to a less radical radicals, hydrogen peroxide (H2O2). As a result of conversion of hydrogen peroxide to water by catalase, peroxidases, or peroxiredoxins, the cell is further preserved from deleterious oxidative stress. On the other hand, SOD2 deficiency leads to superoxide accumulation in the mitochondria, leading to oxidative stress. As both erastin and TMZ treatment produce ROS in cells [13, 32], a depletion of SOD2 in the mitochondria can further increase superoxide lev-els. A higher superoxide level leads to increased lipid peroxidation in membranes and the occurrence of ferroptosis [34]. In an essence, mitochondrial oxidative stress result-ing from SOD2 depletion can facilitate ferroptosis and increase the sensitivity of glioma cells to erastin or TMZ.

Updated Reference Section, please page 21, line 816-823.

  1. Paku, M.; Haraguchi, N.; Takeda, M.; Fujino, S.; Ogino, T.; Takahashi, H.; Miyoshi, N.; Uemura, M.; Mizushima, T.; Yamamoto, H.; Doki, Y.; Eguchi, H., SIRT3-Mediated SOD2 and PGC-1α Contribute to Chemoresistance in Colorectal Cancer Cells. Annals of Surgical Oncology 2021, 28, (8), 4720-4732.
  2. Savic, D.; Steinbichler, T. B.; Ingruber, J.; Negro, G.; Aschenbrenner, B.; Riechelmann, H.; Ganswindt, U.; Skvortsov, S.; Dudás, J.; Skvortsova, I.-I., Erk1/2-Dependent HNSCC Cell Susceptibility to Erastin-Induced Ferroptosis. In Cells, 2023; Vol. 12.
  3. Amos, A.; Jiang, N.; Zong, D.; Gu, J.; Zhou, J.; Yin, L.; He, X.; Xu, Y.; Wu, L., Depletion of SOD2 enhances nasopharyngeal carcinoma cell radiosensitivity via ferroptosis induction modulated by DHODH inhibition. BMC Cancer 2023, 23, (1), 117.
  4. Fujii, J.; Homma, T.; Osaki, T., Superoxide Radicals in the Execution of Cell Death. In Antioxidants, 2022; Vol. 11.

  1. The way how statistical significance is presented on the graphs is not satisfactory. Very often it is not clear which bars/ data points are actually compared. For example, four bars are compared with four bars in figure 3G, which is not correct and not explanatory. Figure 4F – which difference is statistically significant? The same with data quantification in Figure 5B, 5D, 5F, 6D.

Answer: We thank the editor for pointing out this sharp observation. According to this comment, we then revised and improved the clearness of statistical significance to those mentioned figures. Therefore, please kindly refer to our Result section.

Updated figure 4F, please page 9, line 305.

Updated figure 5B, please page 10, line 339.

Updated figure 5D, please page 10, line 339.

Updated figure 5F, please page 10, line 339.

Updated figure 6D, please page 12, line 394.

  1. The figure quality is sometimes low (Figure 1), thus figure text is impossible to decipher. Another issue is straight copy-pasting of the graphs from Excel (?) without further editing, therefore fonts are often disfigured and too small.

Answer: We thank the editor for their very constructive suggestion. According to this suggestion, we then optimized the resolution of figure 1 to increase readability of this main figure. Therefore, please kindly refer to our Result section.

Updated Result Section, please see page 4, line 149.

  1. All experiments were done with use of the one and single GBM cell line. The comparison of the results obtained in U87MG and other cell lines would let to determine how general is the effect. Have Authors made any attempt to check CYBB increase and its consequences upon TMZ treatment in other cell lines? Also, U87MG is described as “mesenchymal” and it gets even more mesenchymal when treated with TMZ. Therefore, performing experiments on non-mesenchymal cell line would be more appropriate and let to study the mesenchymal shift.

Answer: We appreciate the editor for their very constructive and positive advice. Indeed, other reviewer also raised this question and suggestion to reconfirm the finding in other cell line. Providing additional cell line may confirm and increase generalizability of the finding. According to this comment, we then provided other mesenchymal and non-mesenchymal glioblastoma cell line along with the key results of CYBB/Nrf2/SOD2 axis contribution to determine TMZ resistance in our supplementary result. Therefore, please kindly refer to our Result section

Updated Result Section, please see page 8, line 280-303.

To comprehend generalizability of CYBB in contributing development of TMZ resistance across GBM cells, several TMZ-resistant GBM cell lines with different mesenchymal properties were then generated and tested. According to gene set enrichment analysis across multiple GBM cell lines (Fig. 3A), cell lines other than U87MG were then selected as representative of other mesenchymal (Hs683) and non-mesenchymal (T98G) GBM cell lines. The protein level of CYBB and mesenchymal markers expression were then compared between U87MG, Hs683, and T98G cells (Fig. S2A). Both U87MG and Hs683 cells displayed relatively higher mesenchymal markers such as CD44, Vimentin, and N-cadherin along with CYBB than T98G, suggesting high mesenchymal activation in both cells, whereas lacking of mesenchymal marker upregulation in T98G cells denoted non-mesenchymal properties of this cell. Next, Hs683 and T98G GBM cell lines were serially exposed to TMZ similar to the previous step in generating U87MG-R cells. Both Hs683-R and T98G-R exhibited resistance to TMZ treatment than their respective parental cell lines (Fig. S2B and S2C). In resonance to the finding in U87MG-R cells (Fig. 4B), silencing of CYBB re-sensitized both Hs683-R and T98G-R cells upon TMZ treatment (Fig. S2B and S2C). Moreover, knockdown of CYBB repressed Nrf2 and SOD2 expression while deactivating mesenchymal markers such CD44, Slug, Vimentin, and N-Cadherin (Fig. S2D). Interestingly, even in the non-mesenchymal-derived parental GBM cells such as T98G, silencing of CYBB in the TMZ-resistant clones of T98G-R still showed potential effect of repressing NRF2 and SOD2 upregulation while downregulating mesenchymal markers to potentiate tumor suppression of TMZ treatment (Fig. S2C and S2D). Therefore, the data confirmed the general contribution of CYBB in supporting TMZ resistance irrespective to the original properties of mesenchymal phenotype in GBM cells, despite the level of impact might vary between mesenchymal and non-mesenchymal GBM cells.

Updated figure S2, please see supplementary section.

Updated caption figure S2, please see supplementary section.

Figure S2. Role of CYBB in mesenchymal and non-mesenchymal GBM cell lines in the acquisition of TMZ resistance. (A) Western blotting indicated differential expression of CYBB and mesenchymal activation according markers (CD44, Vimentin, N-Cadherin) between presumed mesenchymal cells (U87MG, Hs683) and non-mesenchymal (T98G) GBM cells. The drug response curve showed relative differential sensitivity of TMZ treatment upon CYBB silencing in two chemoresistant GBM cell lines: Hs683 (B) and T98G (C). (D) Western blotting portrayed perturbation of NRF2/SOD2 axis activation and mesenchymal markers (CD44, Slug, Vimentin, N-Cadherin) between presumed Hs683-R and T98G-R chemoresistant GBM cells.

  1. The manuscript title is somehow incomprehensible/misleading. Actually, the provided data show clearly that CYBB promotes both chemoresistance (due to the mesenchymal shift?) and resistance to ferroptosis due to SOD2 activation (not vulnerability to ferroptosis).

Answer: We appreciate the editor for their very positive advice. As per the suggestion, we then revised the main title to reduce any misleading regarding the whole finding of this study. Therefore, please kindly refer to our main Title section.

Updated Title Section, please see page 1, line 2-4.

NADPH Oxidase Subunit CYBB Confers Chemotherapy and Ferroptosis Resistance in Mesenchymal Glioblastoma via Nrf2/SOD2 Modulation

MINOR POINTS:

  1. Figure 4C – X-axis description is incorrect, blue and red bars indicate scramble and shCYBB conditions, respectively.

Answer: We thank the editor for pointing out this sharp observation. As per the suggestion, we then made a correction for this figure 4C description. Therefore, please kindly refer to our Result section.

Updated Result Section, please see figure 4C in page 9, line 305.

Updated caption figure 4C, please see page 9, line 312-313.

Figure 4. (C) Western blotting indicated the downregulation of Nrf2, SOD2, and mesenchymal markers (CD44, Slug, Vimentin, N-Cadherin) upon repression of CYBB.

Updated Result Section, please see page 8, line 265-268.

In addition, the knockdown of CYBB downregulated Nrf2 and SOD2 expression while deactivating expression of several mesenchymal markers such as CD44, Slug, and Vi-mentin in TMZ-resistant U87MG-R cells (Fig. 4C).

  1. Figure 3 is loaded with plenty of information; to make it more comprehensible it could be split into two figures: one figure on CYBB-TMZ-mesenchymal story and second figure on CYBB-Nrf2-SOD2-erastin story. Alternatively, moving to 3F and 3G subpanels to the end of the figure, before mito ROS measurements (subpanel) needs to be considered.

Answer: We appreciate the editor for their suggestion. According to this comment, we then chose the second suggestion to alternatively reconstruct the position and layout of figure 3F-H. As a result, the respective caption figure and result description will be also revised in response to layout reconfiguration. Therefore, please kindly refer to our Result section.

Updated caption figure 3, please see page 7-8, line 249-257.

Figure 3. (F) Immunofluorescence images illustrate the comparative expression of the mitochondrial ROS marker MitoSOX and the mitochondrial label MitoTracker between U87MG and U87MG-R cells. Bar chart quantification of the mean fluorescence intensity of the corresponding marker and cell lines is presented. (G) Drug–response curve represents different sensitivity to erastin in U87MG and U87MG-R. (H) Bar chart compares the percentage of cell death, as shown by cells stained with propidium iodide following the induction of ferroptosis by treatment with either 150 µM tert-butyl hydroperoxide (TBHP) or 10 µM erastin with or without ferroptosis inhibitors (5 mM N-acetyl cysteine [NAC] or 2 µM SRS11-92) between U87MG and U87MG-R.

Updated Result Section, please see page 6, line 220-230.

Here, TMZ-resistant cells exhibited higher mitochondrial oxidative stress along with higher mitochondrial mass, as indicated by the higher fluorescence intensity of Mito-SOX and MitoTracker in U87MG-R cells than in U87MG cells (Fig. 3F). Moreover, TMZ-resistant U87MG-R cells exhibited higher resilience against erastin than did pa-rental U87MG cells (Fig. 3G). PI staining assay was performed to determine the num-ber of dead cells following ferroptosis induction (erastin and TBHP) with or without cotreatment with ferroptosis inhibitors (SRS1192 and NAC). We observed a greater re-duction in the ferroptotic cell death percentage in U87MG-R cells than in parental cells following the induction of ferroptosis with or without cotreatment with a ferroptosis inhibitor, suggesting the attenuation of ferroptosis-specific cell death sensitivity in TMZ-resistant mesenchymal GBM cells (Fig. 3H).

Updated figure layout, please see figure 3 in page 9, line 236.

  1. Figure 2C, 2D – what is actually shown in the Kaplan-Meyer curve graphs: overall survival or progression free survival?

Answer: We thank the editor for their astute observation. Both Kaplan-Meier curves showed and compared differential Progression-Free Survival (PFS) according to different subset of GBM patients. To make it clearer, naming of Progression-Free Survival (PFS)  will be provided in the figure caption and result description. Therefore, please kindly refer to our Result section.

Updated Result Section, please see page 6, line 190-196.

Moreover, in the TCGA–GBM dataset, we observed that high CYBB expression was significantly associated with poor progression-free survival (PFS) in patients with GBM (Fig. 2C). The results of the Kaplan–Meier subgroup analysis by subtype indicat-ed a more significant association between CYBB and lower PFS in mesenchymal GBM cells than in GBM cells of other subtypes (Fig. 2D). However, Cox regression analysis demonstrated that higher CYBB expression was an independent risk factor for signifi-cantly poorer progression-free survival in patients with GBM after the adjustment for subtypes (hazard risk = 1.6; 95% confidence interval = 1.05–2.4; p=0.029) (Fig. 2E).

Updated figure caption of fig 2C-D, please see page 6, line 183-189.

Figure 2. (C) Kaplan–Meier curve presents the association between CYBB expression and progression-free survival (PFS) of patients with GBM in the TCGA-GBM dataset. (D) Subgroup analysis of Kaplan–Meier curves indicated the presence of differential PFS between high and low CYBB expression groups based on annotated subtypes. (E) Forest plot depicts the adjusted hazard ratio of CYBB expression and GBM subtypes determined by performing multivariate Cox regression analysis in accordance to PFS of GBM cohort in TCGA dataset.

  1. Whenever abbreviation appears in the text, the full name should be provided (e.g. for CYBB, Nrf2, PTGS2, TMU-SHH and so on).

Answer: We appreciate the editor for their suggestion. As per the suggestion, we then added full name of each abbreviation in the main text whenever the abbreviation firstly placed. In addition, abbreviation list is also provided. Therefore, please kindly refer to our Introduction and Result section.

Updated full name for each abbreviation please refer to indicated description as below:

  • Cytochrome B-245 Beta Chain (CYBB), please refer to page 3, line 104-105
  • Nuclear factor E2-related factor 2 (Nrf2), please refer to page 3, line 76
  • Prostaglandin Sintase-2 (PTGS2), please refer to page 11, line 368
  • Taipei Medical University-Shuang Ho Hospital (TMU-SHH), please refer to page 6, line 180

Updated abbreviation list, please see page 20, line 748-752.

Abbreviations:

GBM: Glioblastoma; CYBB: Cytochrome B-245 Beta Chain; NRF2: Nuclear factor E2-related factor 2; TMZ: Temozolomide; PTGS2: Prostaglandin Sintase-2; TMU-SHH: Taipei Medical University-Shuang Ho Hospital; SOD2: Superoxide Dismutase 2; ROS: Reactive oxygen species; NOX: NADPH Oxidase.

  1. Line 91: “promoted the generation of glutathione to confer the resistance” instead of “promoted and generated glutathione…”

Answer: We thank the editor for pointing out this sharp observation. As per the suggestion, we then revised the sentence to minimize misunderstandings. Therefore, please kindly refer to our Introduction section.

Updated Introduction Section, please see page 2, line 96-97.

In addition, TMZ treatment promoted the generation glutathione to confer resistance [16].

  1. Line 92: sorafenib is primarily an RTK inhibitor, not ferroptosis inducer

Answer: We thank the editor for their positive advice. Indeed sorafenib is a multityrosine kinase inhibitors that later known to potentially trigger ferroptosis. As per the suggestion, we then revised the sentence to reduce misinterpretations. Therefore, please kindly refer to our Introduction section.

Updated Introduction Section, please see page 2, line 97-98.

Originally described as a multityrosine kinase inhibitor, sorafenib triggered ferroptosis and altered TMZ sensitivity but could be reversed through ROS elimination [17].

  1. Line 299: “… cells originating from tumor components were more than those from peritumoral or…” what actually means that “cells were more”? Please, reformulate this sentence.

Answer: We thank the editor for pointing out this sharp observation. The means of this sentence is actually: “tumor components were more abundant than peritumoral or normal counterparts”. As per the suggestion, we then revised this sentence to reduce misinterpretations. Therefore, please kindly refer to our Result section.

Updated Result Section, please see page 3, line 117-118.

In this dataset, cells originating from tumor components were more abundant than those from peritumoral or normal counterparts and clustered into 14 subgroups (Fig. 1A).

Round 2

Reviewer 1 Report

The authors have adequately and appropriately addressed/answered my comments/concerns for the original manuscript. Therefore, i am happy for this manuscript to be accepted.

Author Response

Dear Reviewer,

My coauthors and I are very appreciative for the constructive, critical, and encouraging comments that were provided by the reviewer on this manuscript. It has been very helpful to have comments that are thorough and useful for improving the manuscript. By many folds, we believe the comments and suggestions have enhanced the scientific value of the revised manuscript. All of their suggestions have been taken into account in the revision process. With the suggestion incorporated in the manuscript, we are submitting a corrected version. We have revised the manuscript in response to the reviewer's comments, and we have provided the following responses:

Response to Reviewers:

Reviewer #1: The authors have adequately and appropriately addressed/answered my comments/concerns for the original manuscript. Therefore, I am happy for this manuscript to be accepted.

Answer: First and foremost, we would like to express our sincerest gratitude for your thorough review and valuable feedback on our manuscript. Your insightful comments and concerns have greatly contributed to the improvement of our work, and we are truly appreciative of the time and effort you have invested in this process. We are also delighted to learn that you find our revised manuscript to be satisfactory and that we have adequately and appropriately addressed your comments and concerns. With your recommendation for acceptance, we look forward to sharing our findings with the broader academic community and hope that our research will contribute to the ongoing discourse in our field. Once again, thank you for your invaluable input and your positive assessment of our manuscript.

Reviewer 2 Report

The manuscript quality has been improved, however there are still some minor points that should be addressed .

1. Results section: Please move the chapter describing Figure S2 content ahead of the chapter describing Figure 4.

2.       The following typos/incorrect wording have been found in the text. Please scan the whole text carefully to eliminate other typos.

Line 25: „tend” needs to be replaced with “tending”

Line 33: “mesenchymal subtype GBM” -> “mesenchymal subtype of GBM”

Line 35: “were” –> “remained” 

Line 38: “molecularly” - > omitting this word is advised

Lines 90 – 91: “to accumulate ROS level either the mitochondrial or lipid ROS” – “to accumulate mitochondrial and lipid ROS”

Line 95: “ferroptosis” -> “ferroptosis inducers”

Line 97: “generation glutathione”  -> “generation of glutathione”

Line 98: “but” -> “which” (suggestion)

Lines 121 and 129: “cluster numbers” - > “clusters numbered”

Line 135: “heterogeneity in tumor cells” -> “heterogeneity observed in tumour cells”

Line 138: “GBM cells” -> just “GBM”, as bulk tumour/tumour tissue is analyzed

Line 169: “GBM cells” -> “GBM patients”

Line 171: TMU-SHH appears in the text for the first time

Line 176: “marked” -> “remarkable”

Lines 192-193: “indicated a more significant association between CYBB and lower PFS in mesenchymal GBM cells than in GBM cells” -> “indicated significant correlation between higher CYBB levels and lower PFS in mesenchymal GBM only”; again, patient data are analyzed, not GBM cells, please be aware of it and scan the whole text for this particular kind of error

Line 207: “a IC50” -> “an IC50”

Line 296: “such CD44” -> “such as CD44”

Line 369: “Sintase” -> “Synthase”

Line 417: “can detect” -> “can help in the detection, but also determines”

Line 429: “targeting CYBB downstream” -> „targeting events downstream of CYBB”

Lines 439 and 452 – please explain the abbreviations: phox, NCF1

Line 460: “activates the activity of Nrf2” -> “activates Nrf2”

Line 490: “support previous studies” -> “corroborate with the previous reports”

Line 500: “to a less radical radicals” -> “to a less harmful radical”

Author Response

Reviewer #2: The manuscript quality has been improved; however, there are still some minor points that should be addressed.

  1. Results section: Please move the chapter describing Figure S2 content ahead of the chapter describing Figure 4.

Answer: We appreciate the reviewer for their suggestion. In accordance with the suggestion, we moved the sentence to improve clarity and overall smoothness of the entire story, as per the recommendation. Therefore, please kindly refer to our Result section.

Updated Result Section, please see page 2, line 260-283.

To comprehend generalizability of CYBB in contributing development of TMZ re-sistance across GBM cells, several TMZ-resistant GBM cell lines with different mesen-chymal properties were then generated and tested. According to gene set enrichment analysis across multiple GBM cell lines (Fig. 3A), cell lines other than U87MG were then selected as representative of other mesenchymal (Hs683) and non-mesenchymal (T98G) GBM cell lines. The protein level of CYBB and mesenchymal markers expression were then compared between U87MG, Hs683, and T98G cells (Fig. S2A). Both U87MG and Hs683 cells displayed relatively higher mesenchymal markers such as CD44, Vimentin, and N-cadherin along with CYBB than T98G, suggesting high mesenchymal activation in both cells, whereas lacking of mesenchymal marker upregulation in T98G cells denoted non-mesenchymal properties of this cell. Next, Hs683 and T98G GBM cell lines were serially exposed to TMZ similar to the previous step in generating U87MG-R cells. Both Hs683-R and T98G-R exhibited resistance to TMZ treatment than their respective parental cell lines (Fig. S2B and S2C). Silencing of CYBB via short-hairpin RNA (shRNA)-mediated knockdown re-sensitized both Hs683-R and T98G-R cells upon TMZ treatment (Fig. S2B and S2C). Moreover, knockdown of CYBB repressed Nrf2 and SOD2 expression while deactivating mesenchymal markers such as CD44, Slug, Vimentin, and N-Cadherin (Fig. S2D). Interestingly, even in the non-mesenchymal-derived parental GBM cells such as T98G, silencing of CYBB in the TMZ-resistant clones of T98G-R still showed potential effect of repressing NRF2 and SOD2 upregulation while downregulating mesenchymal markers to potentiate tumor suppression of TMZ treatment (Fig. S2C and S2D). Therefore, the data confirmed the general contribution of CYBB in supporting TMZ resistance irrespective to the original properties of mesenchymal phenotype in GBM cells, despite the level of impact might vary between mesenchymal and non-mesenchymal GBM cells.

  1. The following typos/incorrect wording have been found in the text. Please scan the whole text carefully to eliminate other typos.

Line 25: “tend” needs to be replaced with “tending”

Answer: We thank the reviewer for their very constructive suggestion. As per the suggestion, we then made correction to respective word. Therefore, please kindly refer to our Abstract section.

Updated Abstract Section, please see page 1, line 25.

Glioblastoma multiforme (GBM) is highly heterogeneous disease with mesenchymal subtype tending to exhibit more aggressive and multitherapy-resistant features.

Line 33: “mesenchymal subtype GBM” -> “mesenchymal subtype of GBM”

Answer: We thank the reviewer for their constructive advice. As per the suggestion, we then made correction to respective word. Therefore, please kindly refer to our Abstract section.

Updated Abstract Section, please see page 1, line 33.

Public transcriptomic data re-analysis found that CYBB and SOD2 were highly upregulated in mesenchymal subtype of GBM.

Line 35: “were” –> “remained”

Answer: We thank the reviewer for their astute observation. As per the suggestion, we then made correction to respective word. Therefore, please kindly refer to our Abstract section.

Updated Abstract Section, please see page 1, line 35.

In vitro study demonstrated that TMZ-resistant GBM cells displayed mesenchymal and stemness features while remained resilient to erastin-mediated ferroptosis by activating CYBB/Nrf2/SOD2 axis.

Line 38: “molecularly” - > omitting this word is advised

Answer: We appreciate the reviewer for their very positive advice. As per the suggestion, we then made correction to respective word. Therefore, please kindly refer to our Abstract section.

Updated Abstract Section, please see page 1, line 38.

Mechanistically, CYBB interacted with Nrf2 and consequently regulated SOD2 transcription.

Lines 90 – 91: “to accumulate ROS level either the mitochondrial or lipid ROS” – “to accumulate mitochondrial and lipid ROS”

Answer: It was a pleasure to receive the reviewer's thoughtful advice. As per the suggestion, we then made correction to respective word. Therefore, please kindly refer to our Introduction section.

Updated Introduction Section, please see page 2, line 90-91.

TMZ treatment has been known to accumulate mitochondrial and lipid ROS, increase labile iron pool, while depleting anti-oxidative capacity in glioma cells [12, 13];

Line 95: “ferroptosis” -> “ferroptosis inducers”

Answer: We appreciate the reviewer for this constructive suggestion. As per the suggestion, we then made correction to respective word. Therefore, please kindly refer to our Introduction section.

Updated Introduction Section, please see page 2, line 95.

Establishing this link may enable the use of ferroptosis inducers to treat gliomas.

Line 97: “generation glutathione”  -> “generation of glutathione”

Answer: We appreciate the reviewer for this positive advice. As per the suggestion, we then made correction to respective word. Therefore, please kindly refer to our Introduction section.

Updated Introduction Section, please see page 2, line 97.

In addition, TMZ treatment promoted the generation of glutathione to confer resistance [16].

Line 98: “but” -> “which” (suggestion)

Answer: We thank the reviewer for pointing out this observation. As per the suggestion, we then made correction to respective word. Therefore, please kindly refer to our Introduction section.

Updated Introduction Section, please see page 2, line 97-99.

Originally described as a multityrosine kinase inhibitor, sorafenib triggered ferroptosis and altered TMZ sensitivity which could be reversed through ROS elimination [17].

Lines 121 and 129: “cluster numbers” - > “clusters numbered”

Answer: We thank the reviewer for this constructive advice. As per the suggestion, we then made correction to respective word. Therefore, please kindly refer to our Result section.

Updated Result Section, please see page 3, line 121.

Individual t-SNE plots were constructed to illustrate the expression levels of our genes of interest, namely CYBB, NFE2L2 (Nrf2), and SOD2, in each cluster; localization of CYBB expression was noted in clusters numbered 5, 7 8, and 13 (Fig. 1B).

Updated Result Section, please see page 3, line 129.

In addition, a comparative analysis of gene signatures indicated that the mesenchymal phenotype, Nrf2 signaling, and treatment resistance were highly activated in previous-ly defined clusters with high CYBB expression, including clusters numbered 7, 8, and 13 (Fig. 1E).

Line 135: “heterogeneity in tumor cells” -> “heterogeneity observed in tumour cells”

Answer: We thank the reviewer for their positive advice. As per the suggestion, we then made correction to respective word. Therefore, please kindly refer to our Result section.

Updated Result Section, please see page 3, line 135.

To confirm intratumor heterogeneity observed in tumor cells at the single-cell lev-el, on the basis of high CYBB expression observed in mesenchymal cells, we examined the transcriptomic profiles of bulk tumor cells.

Line 138: “GBM cells” -> just “GBM”, as bulk tumour/tumour tissue is analyzed

Answer: We thank the reviewer for their astute observation. As per the suggestion, we then made correction to respective word. Therefore, please kindly refer to our Result section.

Updated Result Section, please see page 3, line 138.

We analyzed TCGA GBM dataset and determined that CYBB and SOD2 were highly upregulated in mesenchymal GBM (Fig. 1F).

Line 169: “GBM cells” -> “GBM patients”

Answer: We thank the editor for pointing out this sharp observation. As per the suggestion, we then made correction to respective word. Therefore, please kindly refer to our Result section.

Updated Result Section, please see page 5, line 169.

2.2. CYBB characterized the poor outcomes of mesenchymal GBM patients

Line 171: TMU-SHH appears in the text for the first time

Answer: We appreciate the reviewer for their suggestion. As per the suggestion, we then provide the explanation to the respective abbreviation. Therefore, please kindly refer to our Result section.

Updated Result Section, please see page 5, line 171.

To validate the presence of aberrant CYBB expression in the clinical setting, we examined the tumor specimens of 65 patients with GBM from Taipei Medical University-Shuang Ho Hospital (TMU–SHH).

Line 176: “marked” -> “remarkable”

Answer: We thank the reviewer for this constructive suggestion. As per the suggestion, we then made correction to respective word. Therefore, please kindly refer to our Result section.

Updated Result Section, please see page 5, line 176.

A comparative analysis demonstrated significantly higher expression levels of N-cadherin, CD44, and vimentin in patients with GBM with high CYBB expression, suggesting the remarkable expression of CYBB in mesenchymal GBM cells (Fig. 2B).

Lines 192-193: “indicated a more significant association between CYBB and lower PFS in mesenchymal GBM cells than in GBM cells” -> “indicated significant correlation between higher CYBB levels and lower PFS in mesenchymal GBM only”; again, patient data are analyzed, not GBM cells, please be aware of it and scan the whole text for this particular kind of error

Answer: We thank the reviewer for their constructive advice. As per the suggestion, we then made correction to respective word. Therefore, please kindly refer to our Result section.

Updated Result Section, please see page 6, line 193-194.

The results of the Kaplan–Meier subgroup analysis by subtype indicated significant correlation between CYBB levels and lower PFS in mesenchymal GBM only (Fig. 2D).

Line 207: “a IC50” -> “an IC50”

Answer: We thank the reviewer for pointing out this sharp observation. As per the suggestion, we then made correction to respective word. Therefore, please kindly refer to our Result section.

Updated Result Section, please see page 6, line 208.

To examine TMZ resistance in an in vitro model, we used an approach employed in a previous study with some modifications and generated TMZ-resistant clones of U87MG-R cells with an IC50 higher than that of parental cells (Fig. 3B).

Line 296: “such CD44” -> “such as CD44”

Answer: We thank the reviewer for their positive advice. As per the suggestion, we then made correction to respective word. Therefore, please kindly refer to our Result section.

Updated Result Section, please see page 8, line 276.

Moreover, knockdown of CYBB repressed Nrf2 and SOD2 expression while deactivating mesenchymal markers such as CD44, Slug, Vimentin, and N-Cadherin (Fig. S2D).

Line 369: “Sintase” -> “Synthase”

Answer: We thank the reviewer for their constructive advice. As per the suggestion, we then made correction to respective word. Therefore, please kindly refer to our Result section.

Updated Result Section, please see page 11, line 368.

Knockdown of SOD2 sensitized TMZ-resistant GBM cells to erastin-mediated ferroptosis, as indicated by a depression in the drug–response curve (Fig. 5E), marked suppression of the colony-forming ability (Fig. 5F), and upregulation of the ferroptosis marker Prostaglandin Synthase-2 (PTGS2) (Fig. 5G).

Line 417: “can detect” -> “can help in the detection, but also determines”

Answer: We thank the reviewer for this constructive suggestion. As per the suggestion, we then made correction to respective word. Therefore, please kindly refer to our Discussion section.

Updated Discussion Section, please see page 13, line 417.

We identified a NOX subunit, called CYBB that can help in the detection, but also determines mesenchymal signature activation in GBM cells.

Line 429: “targeting CYBB downstream” -> „targeting events downstream of CYBB”

Answer: We appreciate the reviewer for their suggestion. As per the suggestion, we then made correction to respective word. Therefore, please kindly refer to our Discussion section.

Updated Discussion Section, please see page 13, line 429.

Therefore, targeting events downstream of CYBB may provide an opportunity for clinicians to exploit this specific subpopulation of GBM to provide precision medicine.

Lines 439 and 452 – please explain the abbreviations: phox, NCF1

Answer: We appreciate the reviewer for their very positive advice. As per the suggestion, we then provide explanation to respective abbreviations. Therefore, please kindly refer to our Discussion section.

Updated Discussion Section, please see page 13, line 439.

The membrane-bound subunits CYBB (also called NOX2 or gp91phox, where phox refers to phagocyte oxidase) and p22phox (CYBA) are responsible for the catalytic core of oxidase.

Updated Discussion Section, please see page 13, line 453.

Neutrophil Cytosolic Factor 1 (NCF1) or p47-phox is physically bound to Nrf2, pre-venting the ubiquitination and activation of Nrf2 [21].

Line 460: “activates the activity of Nrf2” -> “activates Nrf2”

Answer: We thank the reviewer for their very constructive suggestion. As per the suggestion, we then made correction to respective word. Therefore, please kindly refer to our Discussion section.

Updated Discussion Section, please see page 14, line 461.

Because of the dual effects of ROS on both resistance and cell death, this study empha-sizes that CYBB-mediated oxidative stress supports the mesenchymal features of GBM, activates Nrf2 to protect GBM cells from high ROS levels, and enhances the resistance of GBM cells to TMZ.

Line 490: “support previous studies” -> “corroborate with the previous reports”

Answer: It was a pleasure to receive the reviewer's thoughtful advice. As per the suggestion, we then made correction to respective word. Therefore, please kindly refer to our Discussion section.

Updated Discussion Section, please see page 14, line 491.

The results of this study corroborate with the previous reports that have highlighted the importance of SOD2 in buffering superoxide accumulation, as well as dictating sensitivity to chemotherapy and pro-ferroptosis agents [7, 25, 26].

Line 500: “to a less radical radicals” -> “to a less harmful radical”

Answer: We thank the reviewer for their positive advice. As per the suggestion, we then made correction to respective word. Therefore, please kindly refer to our Discussion section.

Updated Discussion Section, please see page 14, line 501.

The anti-oxidative function of SOD2 is located within the mitochondrial matrix, where it converts superoxide anions produced by mitochondria during electron transport chain (ETC) to a less harmful radicals, hydrogen peroxide (H2O2).
